# Theoretical Limitations of Ensembles in the Age of Overparameterization

## Abstract

Classic tree-based ensembles generalize better than any single decision tree. In contrast, recent empirical studies find that modern ensembles of (overparameterized) neural networks may not provide any inherent generalization advantage over single but larger neural networks. This paper clarifies how modern overparameterized ensembles differ from their classic underparameterized counterparts, using ensembles of random feature (RF) regressors as a basis for developing theory. In contrast to the underparameterized regime, where ensembling typically induces regularization and increases generalization, we prove that infinite ensembles of overparameterized RF regressors become pointwise equivalent to (single) infinite-width RF regressors. This equivalence, which is exact for ridgeless models and approximate for small ridge penalties, implies that overparameterized ensembles and single large models exhibit nearly identical generalization. As a consequence, we can characterize the predictive variance amongst ensemble members, and demonstrate that it quantifies the expected effects of increasing capacity rather than capturing any conventional notion of uncertainty. Our results challenge common assumptions about the advantages of ensembles in overparameterized settings, prompting a reconsideration of how well intuitions from underparameterized ensembles transfer to deep ensembles and the overparameterized regime.

## 1 Introduction

Ensembling is one of the most well-established techniques in machine learning (e.g. Schapire, 1990; Hansen & Salamon, 1990; Opitz & Maclin, 1999; Dietterich, 2000). Historically, most ensembles aggregated component models that are simple by today's standards. Common techniques like bagging (Breiman, 1996), feature selection (Breiman, 2001), random projections (Kabán, 2014; Thanei et al., 2017), and boosting (Freund, 1995; Chen & Guestrin, 2016) were developed and analyzed assuming decision trees, least-squares regressors, and other *underparameterized* component models incapable of achieving near-zero training error. Crucially, the resulting ensembles achieve better generalization than what could be achieved by any individual component model.

Recently, researchers and practitioners have turned to ensembling large *overparameterized* models, such as neural networks, which have more than enough capacity to memorize training datasets and are typically trained with little to no regularization. Like ensembles of underparameterized models, ensembles of large neural networks are often used to reduce generalization error (Lee et al., 2015; Fort et al., 2019). Motivated by practical effectiveness and heuristics from classic ensembles (Mentch & Hooker, 2016), some have further argued that the predictive variance amongst component models in these so-called *deep ensembles* is a well-calibrated notion of uncertainty (Lakshminarayanan et al., 2017; Ovadia et al., 2019; Gustafsson et al., 2020) that can be used on downstream decision-making tasks (Gal et al., 2017; Yu et al., 2020).

While there are few theoretical works analyzing these modern overparameterized ensembles, recent empirical evidence suggests that intuitions from their underparameterized counterparts do not hold in this new regime. For example, classic methods to increase diversity amongst component models, such as bagging, are harmful for deep ensembles (Nixon et al., 2020; Jeffares et al., 2024; Abe et al., 2024) despite being nearly universally beneficial for underparameterized ensembles. Moreover, several recent studies question whether deep ensembles offer significant improvements in robustness and uncertainty quantification over what can be achieved by a single (but larger) neural network

(Abe et al., 2022; Theisen et al., 2024; Chen et al., 2024). These results suggest that an ensemble of (large) overparameterized networks may not differ fundamentally from a single (extremely large) neural network, in contrast to the underparameterized regime where ensembles are a fundamentally different class of predictors Schapire (1990); Breiman (2001); Kabán (2014).

To address this divergence and verify recent empirical findings on deep ensembles, we develop a theoretical characterization of ensembles in the overparameterized regime, with the goal of contrasting against (traditional) underparameterized ensembles. We answer the following questions:

1. Do large ensembles of overparameterized models differ from single (very large) models trained on the same data? Does the capacity of the component models affect this difference?

2. Under a fixed parameter/computation budget, does an ensemble of overparameterized models provide additional generalization or robustness benefits over a single (larger) model?

3. What does the predictive variance of overparameterized ensembles measure, and does it relate to different notions of uncertainty?

To answer these questions, we analyze ensembles of overparameterized random feature (RF) linear regressors, a widely used theoretically-tractable approximation of neural networks (e.g. Belkin et al., 2018; Bartlett et al., 2020; Mei & Montanari, 2022). These models can be interpreted as neural networks where only the last layer is trained (e.g. Rudi & Rosasco, 2017; Belkin et al., 2019) or as first-order Taylor approximations of neural networks (e.g. Jacot et al., 2018). By averaging models that differ solely in their random features, we emulate the common practice of ensembling neural networks that differ only by random initialization (Lakshminarayanan et al., 2017). Our analysis focuses on the practically relevant regime where regressors are trained with little to no regularization.

## 1.1 RELATED WORK

**Random feature models.** RF models perform regression on a random subset or projection of a high- (or infinite-) dimensional feature representation. Originally introduced as a scalable approximation to kernel machines (Rahimi & Recht, 2007; 2008a;b), RF regressors have seen growing theoretical interest as simplified models of neural networks (e.g. Belkin et al., 2019; Jacot et al., 2018; Bartlett et al., 2020; Mei & Montanari, 2022; Simon et al., 2024). This approximation of neural networks becomes exact in the limit of infinite width (e.g. Jacot et al., 2018; Lee et al., 2019).

**Underparameterized random feature models and ensembles.** There are many works theoretically characterizing ensembles of tree-based models (e.g. Schapire & Singer, 1998; Sexton & Laake, 2009; Wager et al., 2014; Mentch & Hooker, 2016). Here, we restrict our discussion to analyses of (ensembles of) RF regressors. Most works of this nature analyze *underparameterized* models, where the number of random features (i.e., the width) is assumed to be far fewer than the number of data points. In the underparameterized fixed design setting, the infinite ensemble of unregularized RF regressors achieves the same generalization error as ridge regression on the original (unprojected) inputs (Kabán, 2014; Thanei et al., 2017; Bach, 2024a). We emphasize the distinction between underparameterized component models and their aggregated prediction: i.e., the ensemble of unregularized regressors is equivalent to a regularized predictor. (We provide theoretical analysis in Appx. D that further demonstrates ridge-like behaviour of underpameterized RF ensembles.)

**Overparameterized random feature models.** Recent works on RF models have focused on the *overparameterized regime*, often using high-dimensional asymptotics to characterize generalization error (Adlam & Pennington, 2020; Hastie et al., 2022; Mei & Montanari, 2022; Loureiro et al., 2022; Bach, 2024a). Implicit in many works is an assumption of *Gaussian universality*, in which the marginal distributions over the random features are replaced by moment-matched Gaussians. While such assumptions are common throughout asymptotic random matrix theory (e.g. Tao, 2012), our work aims to establish more general results that hold for more general random features. We demonstrate—both theoretically and empirically—that Gaussianity may be an inappropriate approximation for neural network features when comparing the pointwise behaviour of ensembles versus single models.

The benefits of overparameterization and ensembling for out-of-distribution generalization in random feature models have been analyzed by Hao et al. (2024), who provide lower bounds on OOD

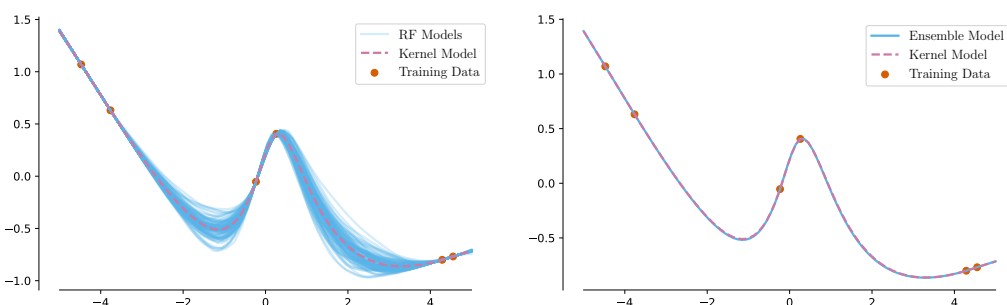

Figure 1: **An infinite ensemble of overparameterized RF models is equivalent to a single infinite-width RF model.** (Left) We show a sample of 100 finite-width RF models (blue) with ReLU activations trained on the same $N = 6$ data points. Additionally, we show the single infinite-width RF model (pink). The finite-width predictions concentrate around the infinite-width model. (Right) We again show the single infinite-width RF model (pink) and the "infinite" ensemble of $M = 10,000$ RF models (blue). We note no perceptible difference between the two.

risk improvements when increasing capacity or using ensembles. Their work focuses on non-asymptotic guarantees under specific distributional shifts, while ours examines the (asymptotic) equivalence of ensembles and single large models under minimal assumptions. Most related to our work is Jacot et al. (2020), who analyze the pointwise expectation and variance of ridge-regularized RF models with Gaussian process (GP) features. We extend this by significantly weakening the assumptions on random features, demonstrating that the convergence of infinite ensembles to infinite-width single models is a general property of overparameterization, independent of Gaussianity or specific feature distributions.

## 1.2 CONTRIBUTIONS

We consider ensembles of *overparameterized* RF regressors in both the ridgeless and small ridge regimes. Unlike prior work, we make minimal assumptions about the distribution of the random features. Therefore, our results can be assumed to hold for most RF ensembles rather than only those that are compositions of GP-random features. Concretely, we make the following contributions:

To answer Question 1: we show that the average ridgeless RF regressor is pointwise equivalent to its corresponding ridgeless kernel regressor (Theorem 1), implying that an infinite ensemble of overparameterized RF models is *exactly* equivalent to a single infinite-width RF model. We further show that this equivalence approximately holds in the small ridge regime (Theorem 2).

To answer Question 2: we use rates established in prior work to demonstrate that the variance reduction from ensembling overparameterized RF regressors is very similar to increasing the number of features in a single model. This shows that ensembles do not offer additional generalization or robustness advantages over single models under fixed parameter budgets (see Sec. 3.2).

To answer Question 3: we show that the predictive variance in an overparameterized ensemble is the expected squared difference between the predictions from a (finite-width) RF regressor and its corresponding kernel regressor (i.e., the infinite-width model). With this finding, we demonstrate that ensemble variance differs from conventional uncertainty quantifications, except in practically unrealistic cases where the random features are sampled from a GP (see Sec. 3.2).

Altogether, these results support recent empirical findings that deep ensembles offer few generalization and uncertainty quantification benefits over single models (Abe et al., 2022; Theisen et al., 2024). Our theory and experiments demonstrate that these phenomena are not specific to neural networks but are more general properties of ensembles in the overparameterized regime.

## 2 SETUP

We work in a regression setting. The training dataset $\mathcal{D} = \{(x_i, y_i)\}_{i=1}^N \in (\mathcal{X} \times \mathbb{R})^N$ is assumed to be a fixed set of size $N$. The vector $y \in \mathbb{R}^N$ represents the concatenation of all training responses.

We consider *RF models* adhering to the form $h_{\mathcal{W}}(x) = \frac{1}{\sqrt{D}} \sum_{i=1}^D \phi(\omega_i, x)\theta_i$, where $\theta_i$ are learned parameters, $\mathcal{W} = \{\omega_i\}_{i=1}^D \in \Omega^D$ are i.i.d. draws from some distribution $\pi(\cdot)$, and $\phi : \Omega \times \mathcal{X} \to \mathbb{R}$ is a *feature extraction function*. In the case of a ReLU-based RF model with $p$-dimensional inputs, we have $\mathcal{X} = \Omega = \mathbb{R}^p$ and $\phi(\omega_i, x) = \max(0, \omega_i^\top x)$. Though RF models cannot fully explain the behaviour of neural networks (e.g. Ghorbani et al., 2019; Li et al., 2021; Pleiss & Cunningham, 2021), they can be a useful proxy for understanding the effects of overparameterization and capacity on generalization (e.g. Belkin et al., 2019; Adlam & Pennington, 2020; Mallinar et al., 2022).

**Notation.** For any $x, x' \in \mathcal{X}$, let $k(x, x') = \mathbb{E}_\omega[\phi(\omega, x)\phi(\omega, x')]$ denote the second moment of the feature extraction function $\phi(\omega, \cdot)$. We note that the function $k$ is a positive definite kernel function, and we will refer to it as such. We will use the matrices $K \coloneqq [k(x_i, x_j)]_{ij} \in \mathbb{R}^{N \times N}$ and $\Phi_{\mathcal{W}} \coloneqq [\phi(\omega_j, x_i)]_{ij} \in \mathbb{R}^{N \times D}$ to denote the kernel function applied to all training data pairs and the feature extraction function applied to all data/feature combinations, respectively. We will drop the subscript $\mathcal{W}$ when the set of random features is clear from context. We assume that $K$ is invertible.

Throughout our analysis, it will be useful to consider the "whitened" feature matrix $W = R^{-\top}\Phi \in \mathbb{R}^{N \times D}$ where $R^\top R = K$ is the Cholesky decomposition of the kernel matrix $K$. When considering a test point $x^* \in \mathcal{X}$ (or equivalently a set of test points), we extend the $K, R, \Phi, W$ notation by

$$\begin{bmatrix} K & [k(x_i, x^*)]_i \\ [k(x^*, x_j)]_j & k(x^*, x^*) \end{bmatrix} = \begin{bmatrix} R & c \\ 0 & r_\perp \end{bmatrix}^\top \begin{bmatrix} R & c \\ 0 & r_\perp \end{bmatrix}, \quad \begin{bmatrix} W \\ w_\perp^\top \end{bmatrix} = \begin{bmatrix} R & c \\ 0 & r_\perp \end{bmatrix}^{-\top} \begin{bmatrix} \Phi \\ [\phi(\omega_i, x^*)]_i \end{bmatrix}. \quad (1)$$

For fixed training/test points, $\mathbb{E}_W[WW^\top] = D \cdot I$, $\mathbb{E}_{w_\perp}[w_\perp^\top w_\perp] = D$ and $\mathbb{E}_{W, w_\perp}[w_\perp^\top W^\top] = 0$ which can be directly derived from $\mathbb{E}_\Phi[\Phi\Phi^\top] = D \cdot K$ (and similar properties for $\phi^*$). Moreover, the columns $[w_i; w_{\perp i}]$ of $[W; w_\perp]$ are i.i.d. since they are transformations of the i.i.d. columns of $\Phi$.

**Overparameterized ridge/ridgeless regressors and ensembles.** As our focus is the overparameterized regime, we assume a computational budget of $D > N$ features ($\mathcal{W} = \{\omega_1, \ldots, \omega_D\} \sim \pi^D$) to construct an RF regressor $h_{\mathcal{W}}(x) = \frac{1}{\sqrt{D}}\phi_{\mathcal{W}}(x)^\top\theta$. We train the regressor parameters $\theta$ to minimize the loss $\|\frac{1}{\sqrt{D}}\Phi_{\mathcal{W}}\theta - y\|_2^2 + \lambda\|\theta\|_2^2$ for some ridge parameter $\lambda \geq 0$. When $\lambda > 0$ this optimization problem admits the closed-form solution $\theta_{\mathcal{W}, \lambda}^{(\mathrm{RR})} = \frac{1}{\sqrt{D}}\Phi_{\mathcal{W}}^\top \left(\frac{1}{D} \cdot \Phi_{\mathcal{W}}\Phi_{\mathcal{W}}^\top + \lambda I\right)^{-1} y$. Although the learning problem is underspecified when $\lambda = 0$ (i.e. in the ridgeless case), the implicit bias of (stochastic) gradient descent initialized at zero leads to the minimum norm interpolating solution $\theta_{\mathcal{W}}^{(\mathrm{LN})} = \frac{1}{\sqrt{D}}(\Phi)^\top \left(\frac{1}{D} \cdot \Phi\Phi^\top\right)^{-1} y$. We denote the resulting ridge(less) regressors as

$$h_{\mathcal{W}}^{(\mathrm{LN})}(\cdot) \coloneqq \frac{1}{\sqrt{D}}\left[\phi(\omega_j, \cdot)\right]_j \theta_{\mathcal{W}}^{(\mathrm{LN})}, \qquad h_{\mathcal{W}, \lambda}^{(\mathrm{RR})}(\cdot) \coloneqq \frac{1}{\sqrt{D}}\left[\phi(\omega_j, \cdot)\right]_j \theta_{\mathcal{W}, \lambda}^{(\mathrm{RR})}.$$

We also consider ensembles of $M$ ridge(less) regressors. We assume that each is trained on a different set of i.i.d. $D > N$ random features $\mathcal{W}_1, \ldots, \mathcal{W}_M \sim \pi^D$ but trained on the same training set. Thus, the only source of randomness in these ensembles comes from the random selection of features $\mathcal{W}_i$, analogous to the standard training procedure of deep ensembles (Lakshminarayanan et al., 2017). The ensemble prediction is given by the arithmetic average of the individual models

$$\bar{h}_{\mathcal{W}_{1:M}}(\cdot) = \frac{1}{M}\sum_{m=1}^M h_{\mathcal{W}_m, \lambda}(\cdot) = \frac{1}{M}\sum_{m=1}^M \left[\frac{1}{D}\left[\phi(\omega_{mj}, \cdot)\right]_j \Phi_{\mathcal{W}_m}^\top \left(\frac{1}{D} \cdot \Phi_{\mathcal{W}_m}\Phi_{\mathcal{W}_m}^\top + \lambda I\right)^{-1}\right] y.$$

**Assumptions.** A key difference between this paper and prior literature is the set of assumptions about the random feature distribution $\pi(\cdot)$. It is commonly assumed that entries in the extended whitened feature matrix $[W; w_\perp]$ are i.i.d. draws from a zero-mean sub-Gaussian distribution (e.g. Bartlett et al., 2020; Bach, 2024b), which implicitly places constraints on $\phi(\cdot, \cdot)$ and $\pi(\cdot)$. Many works further assume *Gaussian universality*—i.e. that the distribution of $W, w_\perp$ can be modeled by i.i.d. standard Gaussian random variables (Adlam & Pennington, 2020; Mei & Montanari, 2022;

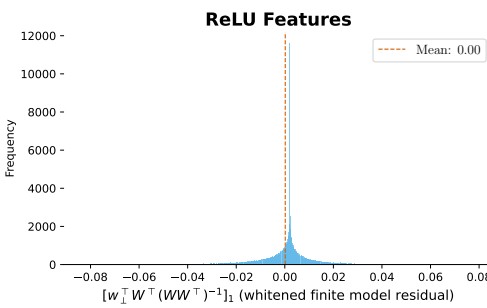 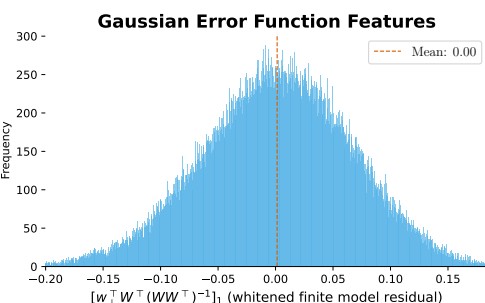

Figure 2: **Empirically, the term** $\mathbb{E}[w_\perp^\top W^\top (WW^\top)^{-1}]$ **is consistently zero.** We plot the distribution of the first index of $w_\perp^\top W^\top (WW^\top)^{-1}$, which captures the difference between the infinite-width single model and a smaller overparameterized RF model (see Eq. (2)). (Left) We use ReLU as activation function, $x_i \in \mathbb{R}$, and $N = 6, D = 200$. (Right) We use the Gaussian Error activation function, the California Housing dataset (Kelley Pace & Barry, 1997), and $N = 12, D = 200$.

Simon et al., 2024)—implying that $\phi(\omega_i, \cdot)$ are draws from a Gaussian process with covariance $k$.[1] We argue this assumption is unrealistic when considering features that resemble those from neural networks. For example, ReLU features are always non-negative and thus the mean of $W = R^{-\top}\Phi$ is almost surely non-zero. Moreover, if $\mathcal{X} \subseteq \mathbb{R}^p$ with $p < N$, then feature extraction functions of the form $\sigma(\omega^\top x)$ are fully specified by a $p$-dimensional random variable. Thus, knowing $N$ evaluations of $\omega_j^\top x_i$ allows one to infer $\omega_j$, making $w_\perp$ deterministic given $W$. We instead consider the following less restrictive assumptions about $W, w_\perp$, which implicitly specify properties of $\pi(\cdot)$:

**Assumption 1** (Assumption of subexponentiality). *We have that*

1. $w_i w_{\perp i}$ *(where $w_i$ is the $i^{\text{th}}$ column of $W$) is sub-exponential $\forall i \in \{1, ..., D\}$ and*

2. $\sum_{i=1}^{D} w_i w_i^\top$ *is almost surely positive definite for any $D \geq N$.*

The first condition is fulfilled whenever $w_{\perp i}$ and $w_i^\top$ are sub-Gaussian (but potentially dependent), which is true when the features come from activation functions with sub-Gaussian weights. The second condition is equivalent to $\Phi$ having almost surely full rank, which is not true for ReLUs and leaky-ReLUs features but which is true for arbitrarily precise approximations thereof.[2] Note we make no assumptions about the mean or independence of the entries in a given column of $[W; w_\perp]$.

## 3 MAIN RESULTS

### 3.1 EQUIVALENCE OF INFINITE ENSEMBLES AND THE INFINITE-WIDTH SINGLE MODELS

We at first assume an infinite computational budget and consider the following two limiting predictors, for which we will show pointwise equivalence in predictions:

1. An infinite-width least norm predictor, $h_\infty^{(\text{LN})}$, the a.s. limit of $h_\mathcal{W}^{(\text{LN})}$ as $|\mathcal{W}| = D \to \infty$

2. An infinite ensemble of finite-width least norm predictors, $\bar{h}_\infty^{(LN)}$, which is the almost sure limit of $\bar{h}_{\mathcal{W}_{1:M}}^{(LN)}$ as $M \to \infty$, with $N < D < \infty$ remaining constant.

These limiting predictors do not only serve as approximations to large ensembles and very large single models but will also prove useful in characterizing the variance and generalization error of finite overparameterized ensembles, as discussed in Sec. 3.2.

---

[1]Assume the entries of $W$ and $w_\perp$ are i.i.d. Gaussian. Then the $i^{\text{th}}$ feature applied to training/test inputs ($[R^\top w_i; c^\top w_i + r_\perp w_{\perp i}]$) is multivariate Gaussian. This fact holds for any train/test data; thus the $i^{\text{th}}$ feature is a GP by definition (e.g. Rasmussen & Williams, 2006, Ch. 2).

[2]E.g., $\phi_\alpha(\omega, x) = \frac{1}{\alpha}\log(1 + e^{\alpha\omega^\top x})$, $\alpha > 0$ yields an a.s. full-rank $\Phi$, and $\phi_\alpha(\omega, x) \overset{\alpha \to \infty}{\to} \text{ReLU}(\omega^\top x)$.

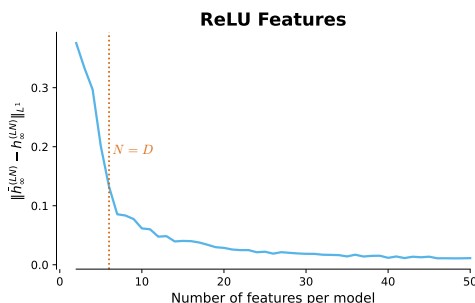
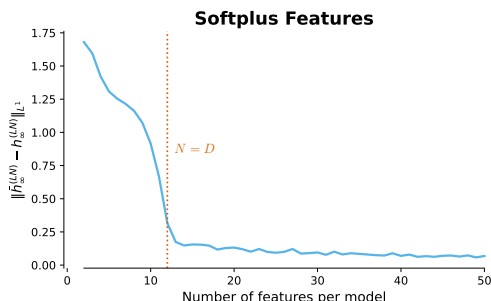

Figure 3: **Infinite overparameterized ensembles are equivalent to a single infinite-width model regardless of width, while underparameterized ensembles behave fundamentally differently.** We present the average absolute difference between the infinite ensemble and the single infinite model for different feature counts $D$. (Left) ReLU activations, $N = 6$, data are from the setting in Fig. 1. (Right) softplus activations, $N = 12$, California Housing dataset. Both exhibit a "hockey stick" pattern: there is a substantial difference between the underparameterized ensemble and the infinite-width model; however, this difference vanishes in the overparameterized regime.

Define $k_N(\cdot) : \mathcal{X} \to \mathbb{R}^N$ as the vector of kernel evaluations with the training data $k_N(\cdot) = [k(x_1, \cdot) \quad \cdots \quad k(x_N, \cdot)]^\top \in \mathbb{R}^N$. As $D \to \infty$, the minimum norm interpolating model converges pointwise almost surely to the ridgeless kernel regressor by the Strong Law of Large Numbers:

$$h_{\mathcal{W}}^{(\mathrm{LN})}(\cdot) \xrightarrow{\text{a.s.}} h_\infty^{(\mathrm{LN})}(\cdot), \qquad h_\infty^{(\mathrm{LN})}(\cdot) := k_N(\cdot)^\top K^{-1} y.$$

On the other hand, using $W$ and $w_\perp$ as introduced in Sec. 2 we can rewrite the infinite ensemble prediction $\bar{h}_\infty^{(LN)}(x^*)$ as (for a derivation of this, see Appx. B.1)

$$\bar{h}_\infty^{(LN)}(x^*) = h_\infty^{(\mathrm{LN})}(x^*) \; + \; r_\perp \mathbb{E}_{W,w_\perp} \left[ w_\perp^\top W^\top \left( WW^\top \right)^{-1} \right] R^{-\top} y \tag{2}$$

To prove the pointwise equivalence of the infinite ensemble and infinite-width single model, we need to show that $\mathbb{E}_{W,w_\perp}[w_\perp^\top W^\top (WW^\top)^{-1}]$ term in Eq. (2) is zero. Note that this result trivially holds when the entries of $W$ and $w_\perp$ are i.i.d., as assumed in prior work (e.g. Jacot et al., 2020). Here, we show that this term is zero even when $w_\perp$ and $W$ are dependent, which—as described in Sec. 2—is a more realistic assumption for neural network features. Empirically, in Fig. 2 we observe that the entries of the random variable $w_\perp^\top W^\top (WW^\top)^{-1} \in \mathbb{R}^N$ have a mean of zero for both ReLU and Gaussian error function features, both of which violate independence assumptions between $w_\perp$ and $W$ (as noted in Sec. 2). We formalize this observation in the following lemma:

**Lemma 1.** *Under Assumption 1, it holds that* $\mathbb{E}_{W,w_\perp}[w_\perp^\top W^\top (WW^\top)^{-1}] = 0$.

*Proof sketch.* (See Appx. B.1 for a full proof.) We start by applying the Woodbury formula to express the matrix inverse $(WW^\top)^{-1}$ as a decomposition involving the matrix $A_{-i} = (WW^\top - w_i w_i^\top)$ and the individual column $w_i$ of $W$. This decomposition yields the expression:

$$w_\perp^\top W^\top (WW^\top)^{-1} = \sum_{i=1}^D (w_{\perp i} w_i^\top)/(1 + w_i^\top A_{-i}^{-1} w_i) A_{-i}$$

Next, using sub-exponential concentration inequalities in conjunction with the Weak Law of Large Numbers, we show that the conditional expectation $\mathbb{E}_{w_{\perp i}, w_i} \left[ (w_{\perp i} w_i^\top)/(1 + w_i^\top A_{-i}^{-1} w_i) \mid A_{-i} \right]$ exists and is zero for all invertible $A_{-i}$. This result implies that: $\mathbb{E}[w_\perp W^\top (WW^\top)^{-1}] = \sum_{i=1}^D \mathbb{E}_{A_{-i}}[\mathbb{E}_{w_{\perp i}, w_i}[(w_{\perp i} w_i^\top)/(1 + w_i^\top A_{-i}^{-1} w_i)] A_{-i}^{-1}] = 0.$ $\qquad \square$

Combining Lemma 1 and Eq. (2) yields the pointwise equivalence of $\bar{h}_\infty^{(LN)}$ and $h_\infty^{(\mathrm{LN})}$:

**Theorem 1** (Equivalence of infinite-width single model and infinite ensembles)**.** *Under Assumption 1, the infinite ensemble of finite-width (but overparameterized) RF regressors $\bar{h}_\infty^{(LN)}$ is pointwise almost surely equivalent to the (single) infinite-width RF regressor $h_\infty^{(\mathrm{LN})}$.*

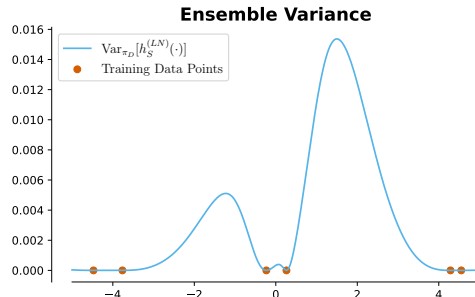 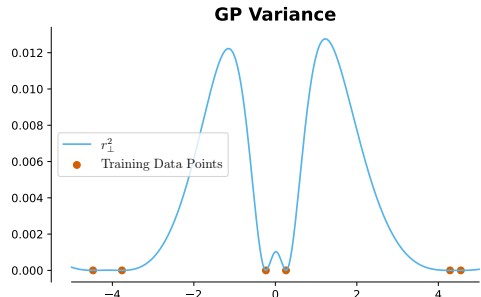

Figure 4: **Ensemble variance (left) and Bayesian notions of uncertainty (right) can differ significantly.** For $N = 6$ and $D = 200$ with ReLU activations, we show the overparameterized ensemble variance (left) and the posterior variance of a Gaussian process with prior covariance $k(\cdot, \cdot)$ (right) across the input range. Empirically, we observe substantial differences between the two quantities.

This result shows that, in overparameterized RF regression, ensembling yields *exactly* the same predictions as simply increasing the capacity of a single model (see Fig. 1 for a visualization). Consequently, we should not expect substantial differences in generalization between large single models and overparameterized ensembles, consistent with recent empirical findings by Abe et al. (2022; 2024); Theisen et al. (2024). Importantly, our result in Theorem 1 holds under minimal assumptions and is independent of Gaussianity, demonstrating that the equivalence between infinite ensembles and infinite-width single models is a fundamental property of overparameterization, applicable to models with non-Gaussian activations and dependent features.

We emphasize a contrast with the underparameterized regime, where RF ensembles match the generalization error of kernel ridge regression (see Appx. D or Bach, 2024a, Sec. 10.2.2). While width controls the implicit ridge parameter in the underparameterized regime (see Sec. 1.1), width does not affect the ensemble predictor in the overparameterized regime. We confirm this difference in Fig. 3 which shows that RF ensembles are equivalent to the ridgeless kernel regressor when $D > N$ but not when $D < N$.

### 3.2 VARIANCE OF ENSEMBLE PREDICTIONS

We now analyze the predictive variance amongst component models in an overparameterized RF ensemble, a quantity used to quantify predictive uncertainty and provide insights about the generalization error. Using Theorem 1, the variance of the predictions of a single RF model with respect to its random features can be expressed as (see Appx. B.2 for a derivation)

$$\text{Var}_{\mathcal{W}}[h_{\mathcal{W}}^{(\text{LN})}(x^*)] = r_\perp^2 \left( y^\top R^{-1} \, \mathbb{E}_{W, w_\perp}[(WW^\top)^{-T} W w_\perp w_\perp^\top W^\top (WW^\top)^{-1}] \, R^{-\top} y \right). \quad (3)$$

In the special case where $W$ and $w_\perp$ are i.i.d. standard normal, this expression simplifies to

$$\text{Var}_{\mathcal{W}}[h_{\mathcal{W}}^{(\text{LN})}(x^*)] = r_\perp^2 \left( \frac{\|h_\infty^{(\text{LN})}\|_k^2}{D - N - 1} \right), \quad (4)$$

where $\|h_\infty^{(\text{LN})}\|_K^2$ represents the squared norm of $h_\infty^{(\text{LN})}$ in the RKHS defined by the kernel $k(\cdot, \cdot)$. From this equation, we note that the variance decreases with RF regressor width as $\sim 1/D$, scales with the complexity of $h_\infty^{(\text{LN})}$, and only depends on $x^*$ through the quantity $r_\perp^2$ (a term we will analyze later).

Unfortunately, $\mathbb{E}_{W, w_\perp}[(WW^\top)^{-T} W w_\perp w_\perp^\top W^\top (WW^\top)^{-1}]$ generally does not have simple expression for arbitrary $W, w_\perp$ satisfying Assumption 1. Without assuming *Gaussian universality*, the variance depends on $x^*$ through both $r_\perp^2$ as well as through the expectation from Eq. (3) involving $w_\perp$. Still, prior works and empirical results (see Fig. 5 and Appx. A.3) suggest that the variance of RF models decays with $\sim 1/D$ under a variety of distributions (e.g. Adlam & Pennington, 2020).

**Implications for uncertainty quantification.** A common approach to uncertainty quantification with ensembles is to examine the predictive variance of their members at a specific test point $x^*$

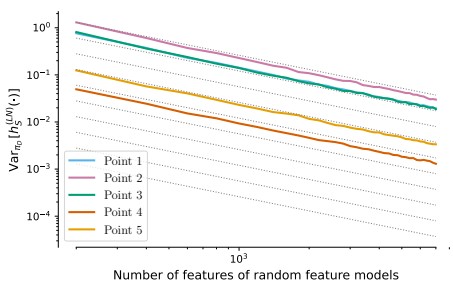 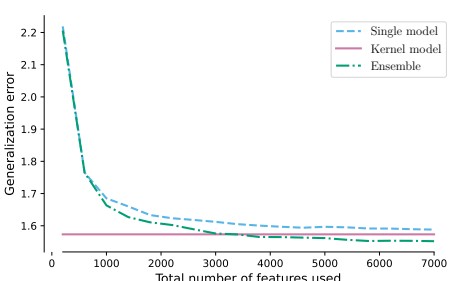

Figure 5: **The variance and generalization error of overparameterized ensembles and single large models scale similarly with the total number of features.** (Left) We show that the variance of a single model with $MD$ features decays as $\sim \frac{1}{MD}$, consistent with the scaling behavior of the variance in an ensemble. (Right) We present the generalization error of an ensemble of $M$ models, each with $D = 200$ features, compared to a single model with $MD$ total features. Both exhibit nearly identical dependence on the total feature budget. The results use a ReLU activation function, the California Housing dataset, and $N = 12$.

(Lakshminarayanan et al., 2017). Before diving into an analysis of Eqs. (3) and (4), it is worth reflecting on the implications that Theorem 1 has for ensemble variance as uncertainty quantification. Because the expected overparameterized RF model is the infinite-width RF model, we can exactly characterize the ensemble variance as the expected squared difference between the predictions of a large (i.e., infinite-width) model versus a smaller (finite-width but still overparameterized) model. This reveals that ensemble variance provides a non-standard notion of uncertainty, differing from both conventional frequentist and Bayesian interpretations.

A notable exception is when $W$ and $w_\perp$ are i.i.d. standard normal. Recall by Eq. (4) that the variance under the *Gaussian universality* assumption only depends on $x^*$ through the quantity $r_\perp^2$. By Eq. (1) we see that $r_\perp^2$ is equal to $k(x^*, x^*) - k_N(x^*)^\top K^{-1} k_N(x^*)$, which is exactly the Gaussian process posterior variance with prior covariance $k(\cdot, \cdot)$ (e.g. Rasmussen & Williams, 2006). In this case, ensembles provide a scaled version of a classic Bayesian estimate of uncertainty.

However, relaxing from Gaussianity to Assumption 1 makes the relationship between ensemble variance and $r_\perp^2$ more complex, as the independence between $W$ and $w_\perp$ is no longer guaranteed. As can be seen in Eq. (3), the variance generally depends on $x^*$ through both $r_\perp^2$ and a complicated expectation involving $W$ and $w_\perp$. In Appx. B.2 we demonstrate with a simple example that this expectation can indeed depend on $x^*$, implying that ensemble variance does not exactly correspond to a scalar multiple of $r_\perp^2$. In our numerical experiments using realistic (i.e., non-Gaussian) random feature distributions, (Fig. 4 and Appx. A.3), we observe significant deviations between the ensemble variance and the Gaussian process posterior variance, further implying that one cannot view ensembles through a classic framework of uncertainty. These discrepancies are particularly important for uncertainty estimation in safety-critical applications or active learning (e.g. Gal et al., 2017; Beluch et al., 2018).

**Ensembles versus larger single models under a finite feature budget.** Our characterization of ensemble variance also holds implications for the generalization error of ensembles versus single models under a finite computational budget. We compare ensembles of $M$ models with $D$ features each ($\bar{h}^{(LN)}_{\mathcal{W}_{1:M}} = \frac{1}{M} \sum_{m=1}^{M} h^{(LN)}_{\mathcal{W}_m}$) to single models with $MD$ features $h^{(LN)}_{\mathcal{W}^*}(\cdot)$ (i.e., here $|\mathcal{W}_m| = D$ for all $m$ and $|\mathcal{W}^*| = MD$). The expected generalization error of either predictor can be decomposed into standard bias and variance terms:

$$\mathbb{E}_h [L(h)] := \mathbb{E}_h [\, \mathbb{E}_x[(h(x) - \mathbb{E}[y \mid x])^2] = L(\mathbb{E}_h[h]) + \mathbb{E}_x [\mathrm{Var}_h(h(x))].$$

Since $h^{(LN)}_{\mathcal{W}^*}$ and $h^{(LN)}_{\mathcal{W}_1}, \ldots, h^{(LN)}_{\mathcal{W}_M}$ share the same expected predictor (as established in Theorem 1), the only difference in the generalization of $h^{(LN)}_{\mathcal{W}^*}$ and $\bar{h}^{(LN)}_{\mathcal{W}_{1:M}}$ arises from their variances. Due to the independence between ensemble members, we have that $\mathrm{Var}_{\mathcal{W}_{1:M}}[\bar{h}^{(LN)}_{\mathcal{W}_{1:M}}(x)] =$

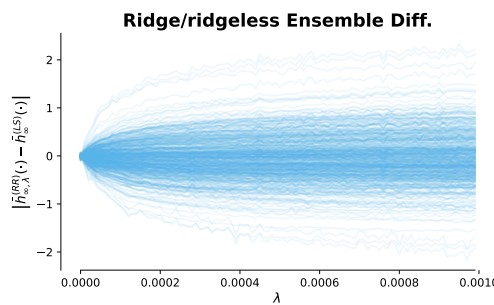 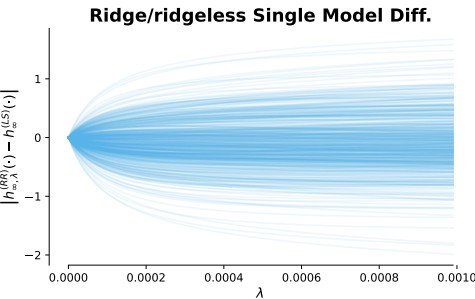

Figure 6: **Lipschitz continuity of predictions for an infinite ensemble and kernel regressor with respect to the ridge parameter.** (Left) We plot the $|\bar{h}_{\infty,\lambda}^{(RR)}(x^*) - \bar{h}_{\infty}^{(LS)}(x^*)|$ as a function of $\lambda$ for 500 test points. (Right) We show the evolution of $|h_{\infty,\lambda}^{(RR)}(x^*) - h_{\infty}^{(LS)}(x^*)|$ for the same test points. Both plots use the ReLU activation function and the California Housing Dataset with $N = 12$ and $D = 200$. While the direct difference $|\bar{h}_{\infty,\lambda}^{(RR)}(x^*) - h_{\infty,\lambda}^{(RR)}(x^*)|$ is not shown due to reasons outlined in Appx. A.4, it can be bounded by a sum of the shown differences (see Appx. C.1).

$\frac{1}{M}\mathrm{Var}_{\mathcal{W}_m}[h_{\mathcal{W}_m}^{(\mathrm{LN})}(x)]$. Moreover, since the variance of a single RF model is inversely proportional to the number of features (exactly in the case of Gaussian features and approximately in the general case, as discussed above), we have that $\mathrm{Var}_{\mathcal{W}^*}[h_{\mathcal{W}^*}^{(\mathrm{LN})}(x)] \, / \, \mathrm{Var}_{\mathcal{W}_m}[h_{\mathcal{W}_m}^{(\mathrm{LN})}] \asymp 1/M$. Altogether, this suggests that the generalization error of finite ensembles and finite-width single models decay at similar rates. We confirm this similar rate of decay in Fig. 5 and Appx. A.3, which compare ensembles versus single models under various feature budgets.

These results further demonstrate that ensembles offer no meaningful generalization advantage over single models. Moreover, since Theorem 1 and the arguments above hold under any test distribution, they align with empirical findings (Abe et al., 2022) that ensembles do not provide robustness benefits beyond those achievable with larger single networks.

### 3.3 EQUIVALENCE OF THE LIMITING PREDICTORS IN THE SMALL RIDGE REGIME

Having established the pointwise equivalence between infinite ensembles and infinite-width single models in the ridgeless regime, we now investigate whether this equivalence approximately persists in the practically relevant setting when a small ridge regularization parameter $\lambda > 0$ is introduced. More generally, we aim to determine whether the transition from the ridgeless case to the small ridge regime is smooth. While $h_{\infty,\lambda}^{(RR)}$, the infinite-width limit of $h_{\mathcal{W},\lambda}^{(RR)}$ as $|\mathcal{W}| = D \to \infty$, almost surely converges to the kernel ridge regressor with ridge $\lambda$, the infinite ensemble $\bar{h}_{\infty,\lambda}^{(RR)} := \mathbb{E}_{\mathcal{W}}[h_{\mathcal{W},\lambda}^{(RR)}(x)]$ does not generally maintain pointwise equivalence with $h_{\infty,\lambda}^{(RR)}$. This divergence occurs even under the *Gaussian universality* assumption (Jacot et al., 2020). However, we hypothesize that the difference between these limiting predictors is small when $\lambda$ is close to zero, which is common in practical applications. To analyze this regime, we introduce a minor additional assumption, which is strictly stronger than the Gaussian universality assumption:

**Assumption 2.** *We assume that $\mathbb{E}_{\mathcal{W}}[(\Phi_{\mathcal{W}}\Phi_{\mathcal{W}}^{\top})^{-1}]$ is finite for all $|\mathcal{W}| = D > N$.*

Under Assumptions 1 and 2, we show that the difference is Lipschitz-continuous with respect to $\lambda$:

**Theorem 2** (The difference between ensembles and large single models is smooth with respect to $\lambda$.)**.** *Under Assumptions 1 and 2, the difference $|\bar{h}_{\infty,\lambda}^{(RR)}(x^*) - h_{\infty,\lambda}^{(RR)}(x^*)|$ between the infinite ensemble and the single infinite-width model trained with ridge $\lambda$ is Lipschitz-continuous in $\lambda$ for $\lambda \geq 0$. The Lipschitz constant is independent of $x^*$ for compact $\mathcal{X}$.*

*Proof sketch.* We first prove a lemma that shows the predictions of infinite-width RF regressors $h_{\infty,\lambda}^{(RR)}(x^*)$ are Lipschitz-continuous in $\lambda$ (see Lemma 3). Using a similar proof strategy and noting

that an equivalent statement to Lemma 1 holds in the ridge regime, we also prove that the predictions of infinite ensembles $\bar{h}_{\infty,\lambda}^{(RR)}(x^*)$ are Lipschitz-continuous in $\lambda$ (see Lemma 4). In Fig. 6, we show how the differences between these predictions and their ridgeless counterparts evolve with respect to $\lambda$ for various test points. Combining these two results and using a triangle inequality yields our theorem. For the full proof, see Appx. C.1. ☐

It is worth noting that even under the stronger assumptions of Jacot et al. (2020), the behavior of the infinite ensemble and the infinite-width model as $\lambda \to 0$ was not fully characterized, as their bounds become vacuous in this limit. Since Theorem 1 ensures that $|\bar{h}_{\infty,\lambda}^{(RR)}(x^*) - h_{\infty,\lambda}^{(RR)}(x^*)| = 0$ for $\lambda = 0$, we can conclude that the pointwise difference grows at most linearly with $\lambda$. Specifically, we have the following bound

$$\left| \bar{h}_{\infty,\lambda}^{(RR)} - h_{\infty,\lambda}^{(RR)}(x) \right| \leq C \cdot \lambda,$$

for some constant $C$ independent of $x^*$, provided that $\mathcal{X}$ is compact. In practical terms, this result indicates that for sufficiently small values of $\lambda$, the predictions of large ensembles and large single models remain nearly indistinguishable, reinforcing our findings from the ridgeless regime.

## 4 CONCLUSION

For *Question 1*, we demonstrated that under weak conditions, infinite ensembles, and single infinite-width models are pointwise equivalent in the ridgeless regime and nearly identical with a small ridge, significantly expanding on prior results (e.g. Jacot et al., 2020). These results verify recent empirical findings (e.g. Abe et al., 2022) that much of the benefit attributed to overparameterized ensembles, such as improved predictive performance and robustness, can be explained by their similarity to larger single models. We contrast these findings to the underparameterized regime, where ensembling typically induces regularization and improves generalization. Similarly, for *Question 2*, we argued that the variance reduction from ensembling is asymptotically equivalent to increasing the number of features of a single model. This result further strengthens our findings on Question 1 and demonstrates functional similarities under relatively small computational budgets. For *Question 3*, we found that the ensemble variance measures the expected difference to a single larger model and is thus a non-standard measure for uncertainty. Significant deviations from the Gaussian process posterior variance indicate that caution is needed when using ensemble variance for uncertainty quantification, especially in safety-critical settings. Again, these results reinforce empirical findings from (Abe et al., 2022) about overparameterized neural network ensembles.

Overall, while our results do not contradict the utility of overparameterized ensembles, they suggest that their benefits may often be explained by their similarity to larger models and that further research is needed to improve uncertainty quantification methods.

**Limitations.** The practical implications of our work are limited by the theoretical abstractions we employ. While these abstractions provide valuable theoretical insights, they may not always hold in real-world, finite settings. Most notably, we approximate neural networks using RF models and focus on infinite single models and infinite ensembles as approximations for large models and large ensembles. Nevertheless, we emphasize that our theoretical results on RF regressors align with recent empirical observations on deep ensembles (Abe et al., 2022; Theisen et al., 2024), further supporting the growing body of work that uses RF models to provide insights into deep learning phenomena (e.g. Belkin et al., 2019; Hastie et al., 2022; Simon et al., 2024).

In addition to these theoretical assumptions, our empirical results are constrained in terms of scale and complexity. Due to numerical stability issues (see Appx. A.2), we primarily considered a small number of samples, a relatively large number of random features, and simple data-generating functions. Again, we refer the readers to the afformentioned empirical work for larger-scale experiments.

## REPRODUCIBILITY AND ETHICS STATEMENTS

**Reproducibility.** The primary contribution of this paper is a theoretical analysis to explain empirical phenomena studied in the recent works of Abe et al. (2022; 2024); Theisen et al. (2024). All

proofs and derivations are largely self-contained, either in the main text or the appendix. We supply references to all background material where applicable.

Empirical results are not the main focus of this work. Nevertheless, we provide the simulation code used to generate all figures in the text, and a complete description of the experiments can be found in Appx. A.1. We also include a discussion on the numerical stability of our experiments in Appx. A.2.

**Ethics.** We believe there are no significant ethical concerns stemming from this work, as it is largely a theoretical analysis of previous empirical results. However, we do note that this work studies ensembles of neural networks and their uncertainty estimates, which have the potential to be used in safety-critical applications (Lakshminarayanan et al., 2017; Ovadia et al., 2019).

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

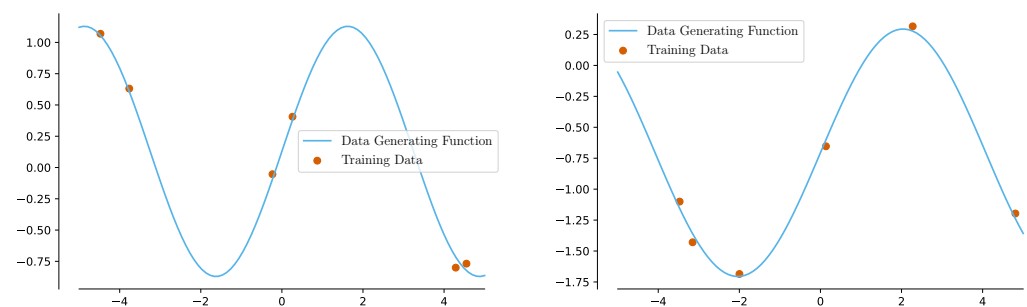

Figure 7: **True function** $f(x) = \sin(5 \cdot b^\top x)$ **with different random seeds.** The blue line shows the true function, while red dots represent training samples for two distinct random seeds.

In the appendix, we will provide the following additional results:

1. In Appx. A, we will describe our experimental setup in more detail, difficulties we encountered when developing the experiments, and provide the results of additional experiments.

2. In Appx. B we will give the proofs for Secs. 3.1 and 3.2 in the main paper.

3. In Appx. C we will give the proofs for Sec. 3.3 in the main paper.

4. Finally, in Appx. D, we prove (under mild assumptions) that infinite underparameterized RF ensembles are equivalent to kernel ridge regression under some transformed kernel.

## A    EXPERIMENTAL SETUP AND ADDITIONAL RESULTS

The code to run all our experiments was attached to the submission. It contains a `README.md` file that explains how to set up and run the experiments.

### A.1   EXPERIMENTAL SETUP

We had two setups using which we performed most of our experiments:

1. We generate training and test points uniformly at random from $[-5, 5]^d$ using the function $f(x) = \sin(5 \cdot b^\top x)$, where $b$ is a vector (depending on the random seed) and the noise parameter is $\sigma = 0.05$ (we assume Gaussian noise with mean 0). In this setting, we use $N = 6, D = 200$, and data from $\mathbb{R}$ (i.e., $d = 1$) if not specified otherwise. You can find a plot of an example true function in Fig. 7.

2. We use the California Housing (Kelley Pace & Barry, 1997) dataset and sample distinct training and test points from it (randomly permutating the dataset initially). In this setting, we use $N = 12, D = 200$ if not differently specified. The data dimension is $\mathbb{R}^8$ here. In contrast to the first setting, we employ a data normalization using a max-min normalization *on the entire dataset* since we experimentally found this makes our methods more stable.

We calculate the generalization error using $N = 1000$ test points in both settings. In the first setting, we calculate the variance of the predictions of a single model using $M = 20,000$ models, while in the second setting, we use $M = 4,000$ models. Apart from Fig. 2 where we use $100,000$ samples, "infinite" ensembles consist of $M = 10,000$ models.

As distribution $\tau(\cdot)$ of the elements $\omega_i \in \mathcal{W}$ we always use $\mathcal{N}(0, I)$. As activation functions, we use ReLU, the Gaussian error function, and the softplus function $\frac{1}{\beta} \cdot \log(1 + \exp(\beta \cdot \omega^\top x))$ with $\beta = 1$. For the first two activation functions, there exist analytically calculatable limiting kernels, the arc-cosine kernel (Cho & Saul, 2009) and the erf-kernel (Williams, 1996). The closed forms for these are

$$k_{\text{arc-cosine}}(x, x') = \frac{1}{2\pi} \|x\| \|x'\| \left( \sin\theta + (\pi - \theta)\cos\theta \right),$$

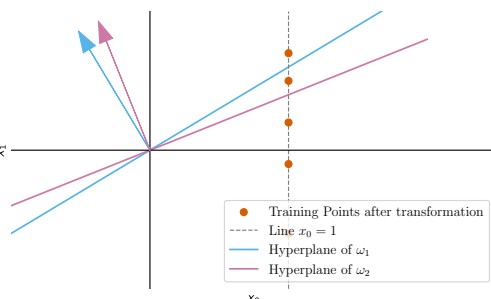

Figure 8: **Visualization of hyperplanes separating training points**. We illustrate how a series of hyperplanes can separate a growing subset of the training points, leading to a triangular, invertible matrix structure as a subset of $\Phi$.

where $\theta = \cos^{-1}\left(\frac{x^\top x'}{\|x\|\|x'\|}\right)$ and

$$k_{\text{erf}}(x, x') = \frac{2}{\pi}\sin^{-1}\left(\frac{2x^\top x'}{\sqrt{(1 + 2\|x\|^2)(1 + 2\|x'\|^2)}}\right).$$

For the softplus function, we approximate the kernel by estimating the second moment $k(x, x') = \mathbb{E}[\phi(\omega, x)\phi(\omega, x') \mid x, x']$ of the feature extraction using $10^7$ samples from $\tau(\cdot)$. For sampling Gaussian features (i.e., testing under the assumption *Gaussian universality*), we use the same approach as described by Jacot et al. (2020).

Before training on data, we always append a $1$ in the zeroeth-dimension of the data before calculating the dot product with $\omega$ (correspondingly, the dimension of $\omega$ is $d + 1$) and applying the activation function. In the ridgeless case, we use $\lambda = 10^{-8}$ to avoid numerical issues.

## A.2 NOTES ON STABILITY

During our experiments, we encountered challenges related to both mathematical stability (i.e., matrices being truly singular rather than nearly singular) and numerical stability. This section outlines these issues and describes the steps we took to mitigate them.

Most importantly, the matrix $\Phi_{\mathcal{W}}\Phi_{\mathcal{W}}^\top$ is not almost surely invertible when using the ReLU activation function, meaning that technically, the second condition of our Assumption 1 is not fulfilled. In numerical experiments, this results in cases where $(\Phi_{\mathcal{W}}\Phi_{\mathcal{W}}^\top)^{-1}$ is nearly singular (though stabilized with $\lambda = 10^{-8}$).

On the other hand, when $D$ is sufficiently large relative to $N$, $\Phi_{\mathcal{W}}$ is full rank with high probability, which implies that $\Phi_{\mathcal{W}}\Phi_{\mathcal{W}}^\top$ is invertible with high probability. Given our data transformation of appending a $1$ in the zeroeth dimension, one can see this as there exists a series of (non-zero probability sets of) hyperplanes separating an increasing subset of the training points, leading to a subset of $\Phi_{\mathcal{W}}$'s columns that form a triangular, invertible matrix (see Fig. 8 for a visualization). Intuitively, higher data dimensionality and better separability of the points increase the probability of $\Phi_{\mathcal{W}}$ having full rank.

As an example of the discussed instabilities, see the adversarial scenario shown in Fig. 9, where $N = 15$ and many training points are placed very close to each other. In this case, individual RF regressors exhibit relatively high variance output values (due to numerical instabilities), which are not averaged out in the "infinite" ensemble. Similar issues were also observed when using the Gaussian error function as the activation function, although they were generally less pronounced.

To alleviate these issues, we used the following approaches:

- We used a relatively low number of samples, $N = 6$ or $N = 12$, compared to $D = 200$. As shown in Fig. 1, even with $D = 200$, there is still a considerable amount of variance in the RF regressors (i.e., the individual RF regressors are not yet closely approximating the limiting kernel ridge regressor).

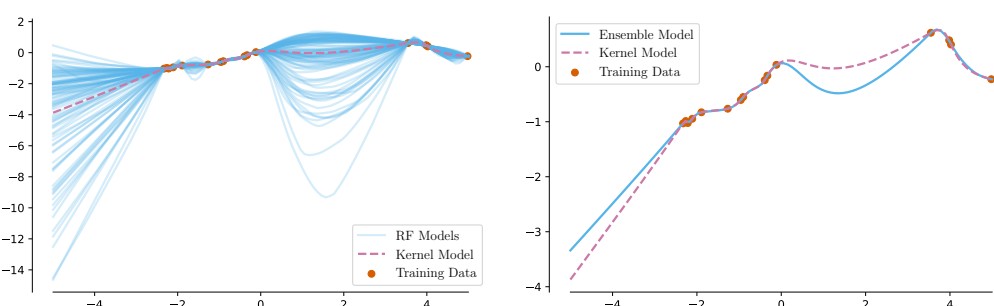

Figure 9: **An adversarial example where the infinite ensemble of overparameterized RF models is numerically not equivalent to a single infinite-width RF model.** (Left) We show a sample of 100 RF models (blue) with ReLU activations trained on the same $N = 15$ densely clustered data points. Additionally, we show the single infinite-width RF model (pink). (Right) We again show the single infinite-width RF model (blue) and the "infinite" ensemble of $M = 10,000$ RF models (pink). A significant difference between the two models is observed in this adversarial case, indicating instability.

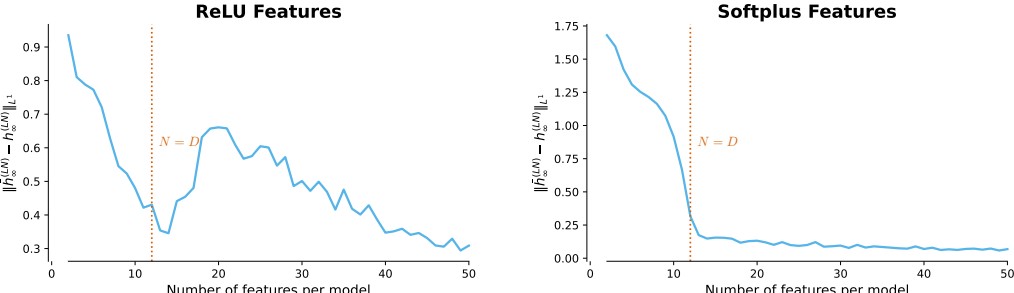

Figure 10: **Using softplus activations instead of ReLU activations reduces instabilities in overparameterized RF ensembles.** The plots show the average absolute difference between the predictions of an infinite ensemble and a single infinite-width model for varying feature counts $D$, using $N = 12$ training samples from the California Housing dataset. (Left) ReLU activations exhibit significant instability, especially for $D > N, D \approx N$, and do not consistently show the expected pointwise equivalence between the infinite ensemble and the single infinite-width model. (Right) Softplus activations — as equivalently shown in Fig. 3 — smooth out these instabilities and more consistently show the expected pointwise equivalence.

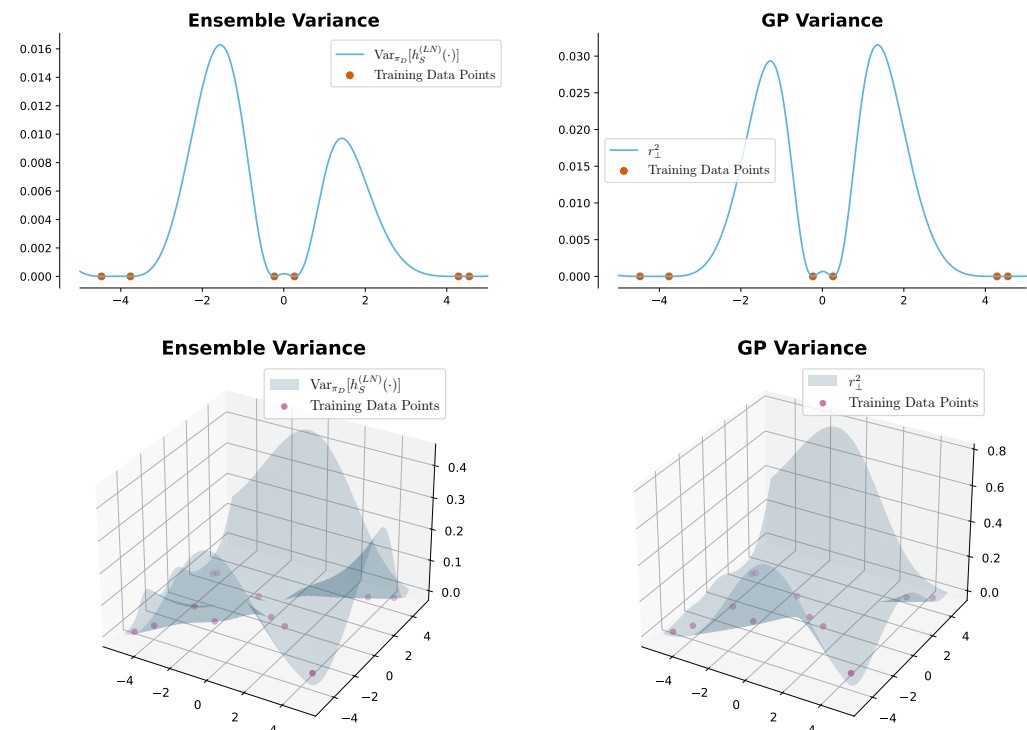

Figure 11: **Variance and $r_\perp^2$ for different activations and dimensions.** (Top left) Variance of RF model predictions across the input range for $D = 200$ and $N = 6$, using the erf activation function. (Top right) Corresponding $r_\perp^2$ values across the input range using the erf kernel. (Bottom left) Variance of RF model predictions across the input range for $D = 200$, $p = 2$, and $N = 12$, using the ReLU activation function. (Bottom right) Corresponding $r_\perp^2$ values across the input range using the arc-cosine kernel.

- We appended a $1$ in the zeroeth dimension of the data before calculating the dot product with $\omega$.
- We performed additional experiments using the softplus function with $\beta = 1$ as a smooth approximation of the ReLU activation function. This often helped stabilize the numerical computations, as seen in Fig. 10, where we repeated a part of the experiment from Fig. 3 using the ReLU function as activation function which increased the numerical instability for low $D$ values.
- We used a ridge term $\lambda = 10^{-8}$ in the ridgeless case to stabilize the inversion of $\Phi_\mathcal{W}\Phi_\mathcal{W}^\top$.
- We used *double precision* for all computations and used the `torch.linalg.lstsq` function with the driver `gelsd` (for not-well-conditioned matrices) to solve linear systems.
- We applied max-min normalization to the entire California Housing dataset to improve stability.

### A.3    ADDITIONAL EXPERIMENTS FOR THE RIDGELESS CASE

**Additional experiments on the ensemble variance.**    We observed a different behavior of the RF regressor variance and $r_\perp^2$ as shown in Fig. 4 consistently across different random seeds and dimensions for both ReLU and the Gaussian error function activations as activation functions. In Fig. 11, we present additional examples for the Gaussian error function in one dimension and the ReLU activation in two dimensions.

**Additional experiment on generalization error and variance scaling.**    In Fig. 5, we demonstrated variance and generalization error decay for the ReLU activation function. To verify the

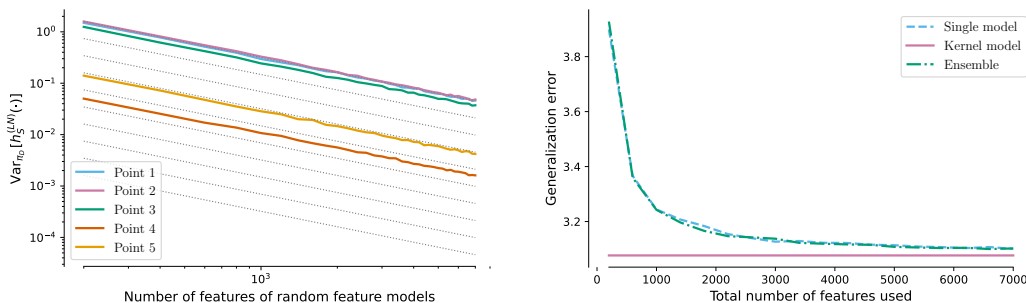

Figure 12: **Variance and generalization error scale similarly with the number of features, consistent with Fig. 5.** In *(a)*, the variance of a single model with $MD$ features decays as $\sim \frac{1}{MD}$, matching the ensemble's behavior. In *(b)*, the generalization error of an ensemble with $M$ models and $D = 200$ features shows a similar decay to that of a single model with $MD$ features. Results use the Gaussian error function, California Housing dataset, and $N = 12$.

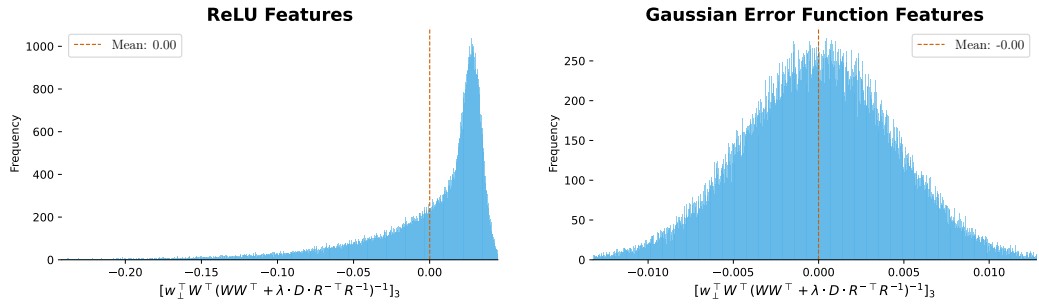

Figure 13: **Empirically, the term** $\mathbb{E}_{W,w_\perp}\left[w_\perp^\top W^\top \left(WW^\top + D \cdot \lambda \cdot R^{-\top}R^{-1}\right)^{-1}\right]$ **is consistently zero.** We show the empirical distribution of the third—just because it looks more interesting—index of $w_\perp^\top W^\top \left(WW^\top + D \cdot \lambda \cdot R^{-\top}R^{-1}\right)^{-1} \in \mathbb{R}^N$, which captures the difference in predictions between $c^\top \mathbb{E}_{W,w_\perp}\left[WW^\top \left(WW^\top + D \cdot \lambda \cdot R^{-\top}R^{-1}\right)^{-1}\right] R^{-\top}y$ and a finite-sized overparameterized RF model (see Eq. (6)). We use $\lambda = 1.0$ in both plots. (Left) We use a ReLU activation function, $x_i \in \mathbb{R}$, and $N = 6, D = 200$. (Right) We use the Gaussian Error Function as activation function, the California Housing dataset, and $N = 12, D = 200$.

consistency of these trends, we repeated the experiment using the Gaussian error function and the corresponding erf-kernel. The results are very similar, shown in Fig. 12.

### A.4 MORE EXPERIMENTS FOR THE RIDGE CASE

**Additional experiments for the convergence of the expected value term.** In Appx. C, we show that a variant of Lemma 1 also holds in the ridge case. More precisely, we show that

$$\mathbb{E}_{W,w_\perp}\left[w_\perp^\top W^\top \left(WW^\top + D \cdot \lambda \cdot R^{-\top}R^{-1}\right)^{-1}\right] = 0$$

under Assumption 1. We repeated the experiment from Fig. 2 for the ridge case to verify this experimentally. The results are shown in Fig. 13.

**Additional notes.** In Fig. 6, we illustrate the Lipschitz continuity of the predictions for an infinite ensemble and a kernel regressor with respect to the ridge parameter. Rather than directly presenting the difference $\left|\bar{h}_{\infty,\lambda}^{(RR)}(x^*) - h_{\infty,\lambda}^{(RR)}(x^*)\right|$, we show the evolution of $\left|\bar{h}_{\infty,\lambda}^{(RR)}(x^*) - \bar{h}_\infty^{(LS)}(x^*)\right|$ and

$\left| h_{\infty,\lambda}^{(RR)}(x^*) - h_{\infty}^{(LS)}(x^*) \right|$. This choice was made because the upper bound we obtained was not consistently tight for settings with large $D$. In particular, the pointwise predictions of the infinite ensemble $\bar{h}_{\infty,\lambda}^{(RR)}$ and the single infinite-width model $h_{\infty,\lambda}^{(RR)}$ trained with ridge $\lambda$ were already very close for non-zero $\lambda$. We opted to display the upper bounds rather than the direct difference to avoid cherry-picking favorable settings.

Our best explanation for this phenomenon is that infinite ensembles under Assumption 1 in the ridge regime often behave similarly to the single infinite-width model $h_{\infty,\tilde{\lambda}}^{(RR)}$ with an *implicit ridge* parameter $\tilde{\lambda}$, which solves the equation

$$\tilde{\lambda} = \lambda + \frac{\tilde{\lambda}}{D} \sum_{i=1}^{N} \frac{d_i}{\tilde{\lambda} + d_i}$$

where $d_i$ are the eigenvalues of the kernel matrix $K$, as shown by Jacot et al. (2020) under *Gaussian universality*. Intuitively and empirically, for large $D$, the implicit ridge $\tilde{\lambda}$ tends to be very close to the true ridge $\lambda$. Using Lemma 3, this suggests that for small values of $\lambda$, the difference between the infinite ensemble and the infinite-width single model $h_{\infty,\lambda}^{(RR)}$ with ridge $\lambda$ is already minimal before $\lambda$ approaches zero.

Interestingly, our findings (see Fig. 3) suggest that in the ridgeless case, the similarity to the ridge regressor with the implicit ridge only holds in the overparameterized regime. Note that this does not violate the results from Jacot et al. (2020) since the constants in their bounds blow up as $\lambda \to 0$ in both the underparameterized and overparameterized regimes.

## B    PROOFS FOR OVERPARAMETERIZED RIDGELESS REGRESSION

### B.1    EQUIVALENCE OF INFINITE ENSEMBLE AND INFINITE SINGLE MODEL.

We start by proving the equivalent formulation of the infinite ensemble prediction stated in Eq. (2) using the terms $W$ and $w_\perp$ as introduced in Sec. 2:

*Proof.* Defining $\phi_{\mathcal{W}}^* = [\phi(\omega_i, x^*)]_i \in \mathbb{R}^D$, we have

$$\bar{h}_\infty(x^*) = \mathbb{E}_{\mathcal{W}} \left[ \frac{1}{D} \phi_{\mathcal{W}}^* \Phi_{\mathcal{W}}^\top \left( \frac{1}{D} \cdot \Phi_{\mathcal{W}} \Phi_{\mathcal{W}}^\top \right)^{-1} \right] y$$

$$= \mathbb{E}_{W,w_\perp} \left[ \left( c^\top W + r_\perp w_\perp^\top \right) W^\top R \left( R^\top W W^\top R \right)^{-1} \right] y$$

$$= \mathbb{E}_{W,w_\perp} \left[ \left( c^\top W + r_\perp w_\perp^\top \right) W^\top \left( W W^\top \right)^{-1} \right] R^{-\top} y$$

$$= c^\top R^{-\top} y \; + \; r_\perp \mathbb{E}_{W,w_\perp} \left[ w_\perp^\top W^\top \left( W W^\top \right)^{-1} \right] R^{-\top} y, \tag{5}$$

where $c, R, r_\perp$ are as defined in Eq. (1). The left term in Eq. (5) is equal to $h_\infty^{(LN)}(x)$:

$$h_\infty^{(LN)}(x^*) = [k(x_i, x^*)]_{i=1}^N K^{-1} y = c^\top R R^{-1} R^{-\top} y = c^\top R^{-\top} y.$$

$\square$

In the case of $\lambda > 0$, we can similarly see that

$$\bar{h}_{\infty,\lambda}^{(RR)}(x^*) = \mathbb{E}_{\mathcal{W}} \left[ \frac{1}{D} \phi_{\mathcal{W}}^* \Phi_{\mathcal{W}}^\top \left( \frac{1}{D} \cdot \Phi_{\mathcal{W}} \Phi_{\mathcal{W}}^\top + \lambda I \right)^{-1} \right] y$$

$$= \mathbb{E}_{W,w_\perp} \left[ \left( c^\top W + r_\perp w_\perp^\top \right) W^\top R \left( R^\top W W^\top R + D \cdot \lambda \cdot R^\top R^{-\top} R^{-1} R \right)^{-1} \right] y$$

$$= \mathbb{E}_{W,w_\perp} \left[ \left( c^\top W + r_\perp w_\perp^\top \right) W^\top \left( W W^\top + D \cdot \lambda \cdot R^{-\top} R^{-1} \right)^{-1} \right] R^{-\top} y$$

$$= c^\top \mathbb{E}_{W,w_\perp} \left[ W W^\top \left( W W^\top + D \cdot \lambda \cdot R^{-\top} R^{-1} \right)^{-1} \right] R^{-\top} y$$

$$+ r_\perp \mathbb{E}_{W,w_\perp} \left[ w_\perp^\top W^\top \left( W W^\top + D \cdot \lambda \cdot R^{-\top} R^{-1} \right)^{-1} \right] R^{-\top} y. \tag{6}$$

Note that the simplification demonstrated in Eq. (2) does not work as nicely in the underparameterized case ($D \leq N$). This is because the weights, in this case, are given by $\theta = (\Phi_{\mathcal{W}}^\top \Phi_{\mathcal{W}})^{-1} \Phi_{\mathcal{W}}^\top y$, and thus the infinite ensemble prediction expands as:

$$\bar{h}_\infty(x^*) = \mathbb{E}_{\mathcal{W}} \left[ \phi_{\mathcal{W}}^* \left( \Phi_{\mathcal{W}}^\top \Phi_{\mathcal{W}} \right)^{-1} \Phi_{\mathcal{W}}^\top \right] y$$

$$= \mathbb{E}_{W, w_\perp} \left[ \left( c^\top W + r_\perp w_\perp^\top \right) \left( W^\top R R^\top W \right)^{-1} W^\top R \right] y.$$

Here, $RR^\top$ lies inside the inverse, preventing the simplifications available in the overparameterized regime.

Next up, we show that the expected value $\mathbb{E}_{W, w_\perp} \left[ w_\perp^\top W^\top \left( WW^\top \right)^{-1} \right]$ is zero under Assumption 1. This directly implies the pointwise equivalence of the infinite ensemble and the single infinite-width model (see Theorem 1).

**Lemma 1 (Restated).** *Under Assumption 1, it holds that* $\mathbb{E}_{W, w_\perp}[w_\perp^\top W^\top (WW^\top)^{-1}] = 0$.

*Proof.* Define $A_{-i} = (WW^\top - w_i w_i^\top)$. Note that $A_{-1}$ is almost surely invertible and positive definite by assumption Assumption 1.

By the Woodbury formula, for almost every $WW^\top$ we have that

$$(WW^\top)^{-1} = (A_{-i} + w_i w_i^\top)^{-1} = A_{-i}^{-1} - \frac{A_{-i}^{-1} w_i w_i^\top A_{-i}^{-1}}{1 + w_i^\top A_{-i}^{-1} w_i},$$

which implies that

$$w_\perp^\top W^\top (WW^\top)^{-1} = \sum_{i=1}^D w_{\perp i} w_i^\top \left( A_{-i}^{-1} - \frac{A_{-i}^{-1} w_i w_i^\top A_{-i}^{-1}}{1 + w_i^\top A_{-i}^{-1} w_i} \right)$$

$$= \sum_{i=1}^D w_{\perp i} \left( w_i^\top \frac{A_{-i}^{-1} + w_i^\top A_{-i}^{-1} w_i^\top A_{-i}^{-1} w_i}{1 + w_i^\top A_{-i}^{-1} w_i} - \frac{w_i^\top A_{-i}^{-1} w_i w_i^\top A_{-i}^{-1}}{1 + w_i^\top A_{-i}^{-1} w_i} \right)$$

$$= \sum_{i=1}^D \frac{w_{\perp i} w_i^\top}{1 + w_i^\top A_{-i}^{-1} w_i} A_{-i}^{-1}.$$

For any positive definite matrix $B \in \mathbb{R}^{N \times N}$ and any vector $v \in \mathbb{R}^N; \|v\| = 1$ and any $i \in \{1, ..., D\}$, we have

$$\left| \mathbb{E}_{w_{\perp i}, w_i} \left[ \frac{w_{\perp i} w_i^\top}{1 + w_i^\top B w_i} \right] v \right| \leq \mathbb{E}_{w_{\perp i}, w_i} \left[ \left| \frac{w_{\perp i} w_i^\top v}{1 + w_i^\top B w_i} \right| \right]$$

$$= \int_0^\infty \mathbb{P} \left[ \left| \frac{w_{\perp i} w_i^\top v}{1 + w_i^\top B w_i} \right| \geq t \right] dt$$

$$= \int_0^\infty \mathbb{P} \left[ \left| w_{\perp i} w_i^\top v \right| \geq \left( 1 + w_i^\top B w_i \right) t \right] dt$$

$$\leq \int_0^\infty \mathbb{P} \left[ \left| w_{\perp i} w_i^\top \right| > t \right] dt$$

$$\leq \int_0^{\nu^2/\alpha} 2 \exp \left( -\frac{t^2}{2\nu} \right) dt + \int_{\nu^2/\alpha}^\infty 2 \exp \left( -\frac{t}{2\alpha} \right) dt, \qquad (7)$$

where the last inequality is a standard sub-exponential bound applied to $w_{\perp i} w_i$. Note that we here use the fact that $\mathbb{E}[w_{\perp i} w_i^\top] = 0$ and the $(\nu^2, \alpha)$-sub-exponentiality of $\left| w_{\perp i} w_i^\top \right|$.

Since the last two integrals in Eq. (7) are finite, the expectation $\mathbb{E}_{W, w_\perp} \left[ (w_{\perp i} w_i^\top)/(1 + w_i^\top B w_i) \right] v$ is finite. By the weak law of large numbers, for i.i.d. random variables $w_i^{(j)}$ and $w_{\perp i}^{(j)}$ across different $j$'s, we have

$$\mathbb{P} \left[ \left| \frac{1}{M} \sum_{j=1}^M \frac{w_{\perp i}^{(j)} (w_i^{(j)})^\top v}{1 + (w_i^{(j)})^\top B w_i^{(j)}} - \mathbb{E}_{W, w_\perp} \left[ \frac{w_{\perp i} w_i^\top}{1 + w_i^\top B w_i} \right] v \right| > t \right] \to 0,$$

for any $t > 0$ and $v \in \mathbb{R}^N$ such that $\|v\| = 1$ as $M \to \infty$. At the same time, repeating the sub-exponential argument above, we have that

$$\mathbb{P}\left[\left|\frac{1}{M}\sum_{j=1}^{M}\frac{w_{\perp i}^{(j)}(w_i^{(j)})^{\top}v}{1+(w_i^{(j)})^{\top}Bw_i^{(j)}}\right| > t\right] \le \mathbb{P}\left[\left|\frac{1}{M}\sum_{i=1}^{M}w_{\perp i}^{(j)}(w_i^{(j)})^{\top}v\right| > t\right]$$

$$\le \begin{cases} 2\exp\left(-\frac{Mt^2}{2\nu}\right) & 0 < t \le \nu^2/\alpha \\ 2\exp\left(-\frac{Mt}{2\alpha}\right) & t > \nu^2/\alpha \end{cases}$$

$$\to 0$$

as $M \to \infty$. Here we use the property that the sum of $M$ $(\nu^2, \alpha)$-sub-exponential random variables is $(M\nu^2, \alpha)$-sub-exponential.

Together, these results imply that $\mathbb{E}_{W,w_{\perp}}\left[(w_{\perp i}w_i^{\top})/(1 + w_i^{\top}Bw_i)\right] = 0$ for every positive definite $B$. Since the random matrix $A_{-i}$ is positive semidefinite, almost surely invertible (by the second half of Assumption 1), and independent of $w_i, w_{\perp i}$, we have that

$$\mathbb{E}_{w_{\perp},W}\left[w_{\perp}W^{\top}\left(WW^{\top}\right)^{-1}\right] = \sum_{i=1}^{D}\mathbb{E}_{w_{\perp i},w_i,A_{-i}}\left[\frac{w_{\perp i}w_i^{\top}}{1+w_i^{\top}A_{-i}^{-1}w_i}A_{-i}^{-1}\right]$$

$$= \sum_{i=1}^{D}\mathbb{E}_{A_{-i}}\left[\mathbb{E}_{w_{\perp i},w_i}\left[\frac{w_{\perp i}w_i^{\top}}{1+w_i^{\top}A_{-i}^{-1}w_i}\right]A_{-i}^{-1}\right] = 0.$$

$\square$

We remark that this proof equivalently holds for the ridge-regression case, i.e., $\mathbb{E}_{W,w_{\perp}}\left[w_{\perp}^{\top}W^{\top}\left(WW^{\top} + D \cdot \lambda \cdot R^{-\top}R^{-1}\right)^{-1}\right] = 0$ since the proof does not rely on the specific form of the matrix $A_{-i}$ other than it being positive definite. Thus by Eq. (6) we directly get that under Assumption 1 it holds that

$$\bar{h}_{\infty,\lambda}^{(RR)}(x^*) = c^{\top}\mathbb{E}_{W,w_{\perp}}\left[WW^{\top}\left(WW^{\top} + D \cdot \lambda \cdot R^{-\top}R^{-1}\right)^{-1}\right]R^{-\top}y. \tag{8}$$

### B.2 Variance of Ensemble Predictions.

In the next step, we show the formula for the variance of a single model prediction under *Gaussian universality*. Note that one could also get this result by slightly extending proofs by Jacot et al. (2020).

**Lemma 2** (Variance of single model predictions). *Under* Gaussian universality *and assuming $D > N + 1$, the variance of single model prediction at a test point $x^*$ is given by*

$$\mathrm{Var}_{\mathcal{W}}[h_{\mathcal{W}}^{(\mathrm{LN})}(x^*)] = r_{\perp}^2\frac{\|h_{\infty}^{(\mathrm{LN})}\|_{\mathcal{H}}^2}{D - N - 1}, \tag{9}$$

*where $\|\cdot\|_{\mathcal{H}}$ is norm defined by the RKHS associated with kernel $k(\cdot, \cdot)$.*

*Proof.* We start by writing down the variance of the prediction of a single model:

$$\mathrm{Var}_{\mathcal{W}}[h_{\mathcal{W}}^{(\mathrm{LN})}(x^*)] = \mathbb{E}_{\mathcal{W}}[h_{\mathcal{W}}^{(\mathrm{LN})}(x^*)^2] - \mathbb{E}_{\mathcal{W}}[h_{\mathcal{W}}^{(\mathrm{LN})}(x^*)]^2$$

Using Theorem 1, the definition of the prediction of a single model and the definition of $W$ and $w_\perp$, we can expand this expression:

$$= \mathbb{E}_{\mathcal{W}}[\phi_{\mathcal{W}}^* \Phi_{\mathcal{W}}^\top (\Phi_{\mathcal{W}} \Phi_{\mathcal{W}}^\top)^{-1} yy^\top (\Phi_{\mathcal{W}} \Phi_{\mathcal{W}}^\top)^{-\top} \Phi_{\mathcal{W}} \phi_{\mathcal{W}}^{*\top}] - (h_\infty^{(\text{LN})}(x^*))^2$$

$$= \mathbb{E}_{W,w_\perp}[(r_\perp w_\perp^\top + c^\top W)W^\top R \left(R^\top WW^\top R\right)^{-1} yy^\top \left(R^\top WW^\top R\right)^{-\top} R^\top W(r_\perp w_\perp^\top + c^\top W)^\top]$$
$$- (h_\infty^{(\text{LN})}(x))^2$$

$$= \mathbb{E}_{W,w_\perp}[(r_\perp w_\perp^\top + c^\top W)W^\top \left(WW^\top\right)^{-1} R^{-\top} yy^\top R^{-1} \left(WW^\top\right)^{-\top} W(r_\perp w_\perp^\top + c^\top W)^\top]$$
$$- (h_\infty^{(\text{LN})}(x))^2$$

$$= (c^\top R^{-\top} y)^2 - (h_\infty^{(\text{LN})}(x))^2$$
$$+ 2 \cdot r_\perp^\top \mathbb{E}_{W,w_\perp}[w_\perp^\top W^\top (WW^\top)^{-1}]R^{-\top} yy^\top R^{-1} c$$
$$+ r_\perp^2 \mathbb{E}_{W,w_\perp}[w_\perp^\top W^\top (WW^\top)^{-1} R^{-\top} yy^\top R^{-1} (WW^\top)^{-T} W w_\perp]$$

Now we can see that the first two terms cancel out (since $h_\infty^{(\text{LN})}(x) = c^\top R^{-\top} y$) and the third term is zero by Lemma 1. We are left with the fourth term, which we can slightly rewrite:

$$\text{Var}_{\mathcal{W}}[h_{\mathcal{W}}^{(\text{LN})}(x^*)] = r_\perp^2 \mathbb{E}_{W,w_\perp}[w_\perp^\top W^\top (WW^\top)^{-1} R^{-\top} yy^\top R^{-1} (WW^\top)^{-T} W w_\perp]$$
$$= r_\perp^2 y^\top R^{-1} \mathbb{E}_{W,w_\perp}[(WW^\top)^{-T} W w_\perp w_\perp^\top W^\top (WW^\top)^{-1}] R^{-\top} y \qquad (10)$$

Using the tower rule for conditional expectations, we have:

$$\text{Var}_{\mathcal{W}}[h_{\mathcal{W}}^{(\text{LN})}(x)] = r_\perp^2 y^\top R^{-1} \mathbb{E}_{W,w_\perp}[(WW^\top)^{-T} W w_\perp w_\perp^\top W^\top (WW^\top)^{-1}] R^{-\top} y$$
$$= r_\perp^2 y^\top R^{-1} \mathbb{E}_W[(WW^\top)^{-T} W \mathbb{E}_{w_\perp | W}[w_\perp w_\perp^\top | W] W^\top (WW^\top)^{-1}] R^{-\top} y$$

Since the *Gaussian universality* assumption implies $W$ and $w_\perp$ are independent, we get:

$$\text{Var}_{\mathcal{W}}[h_{\mathcal{W}}^{(\text{LN})}(x)] = r_\perp^2 y^\top R^{-1} \mathbb{E}_W[(WW^\top)^{-T} W \mathbb{E}_{w_\perp}[w_\perp w_\perp^\top] W^\top (WW^\top)^{-1}] R^{-\top} y$$

Moreover, since by *Gaussian universality* $w_\perp$ and $W$ are multivariate Gaussians with the identity matrix as covariance, we get (via the expected value of a Wishart and an inverse Wishart distribution; note that for getting this expected value, we need to assume that $D > N + 1$):

$$\text{Var}_{\mathcal{W}}[h_{\mathcal{W}}^{(\text{LN})}(x)] = r_\perp^2 y^\top R^{-1} \mathbb{E}_W[(WW^\top)^{-T} (WW^\top)(WW^\top)^{-1}] R^{-\top} y$$
$$= r_\perp^2 y^\top R^{-1} \mathbb{E}_W[(WW^\top)^{-T}] R^{-\top} y$$
$$= r_\perp^2 \frac{y^\top R^{-1} R^{-\top} y}{D - N - 1}$$
$$= r_\perp^2 \frac{y^\top K^{-1} y}{D - N - 1}.$$

Recognizing that $y^\top K^{-1} y = \|h_\infty^{(\text{LN})}\|_{\mathcal{H}}^2$ (e.g. Wainwright, 2019, Ch. 12) completes the proof. $\square$

An equivalent argument does not work under the more general Assumption 1 since $w_\perp$ and $W$ are not necessarily independent. Even in the case of independence, $\mathbb{E}_W[(WW^\top)^{-1}]$ might not be known.

**Counterexample for subexponential case.** We now give an explicit counterexample showing that when only assuming uncorrelatedness between $W$ and $w_\perp$ the term

$$E := \mathbb{E}_{W,w_\perp}[(WW^\top)^{-T} W w_\perp w_\perp^\top W^\top (WW^\top)^{-1}]$$

from Eq. (10) depends on $x^*$ implying that the variance does not only depend on $x^*$ via $r_\perp^2$.

Let us assume $N = D = 1$ and let $W$ be uniformly distributed across the set $\left\{-\frac{4}{\sqrt{12.5}}, -\frac{3}{\sqrt{12.5}}, \frac{3}{\sqrt{12.5}}, \frac{4}{\sqrt{12.5}}\right\}$. Then we have $\mathbb{E}[W] = 0$ and $\mathbb{E}[W^2] = \frac{1}{2} \cdot \frac{16}{12.5} + \frac{1}{2} \cdot \frac{9}{12.5} = 1$.

Now consider an $x^*$ that produces a $w_\perp$ so that $w_\perp = \sqrt{2}$ when $W = \left\{-\frac{3}{\sqrt{12.5}}, \frac{3}{\sqrt{12.5}}\right\}$ and $w_\perp = 0$ otherwise. Then we have $\mathbb{E}[w_\perp^\top W] = 0$ and $\mathbb{E}[w_\perp^2] = 1$. The value of $E$ is now $\frac{12.5}{9}$.

Furthermore, consider an $x^*$ that produces a $w_\perp$ so that $w_\perp = \sqrt{2}$ when $W = \left\{-\frac{4}{\sqrt{12.5}}, \frac{4}{\sqrt{12.5}}\right\}$ and $w_\perp = 0$ otherwise. Then we have $\mathbb{E}[w_\perp^\top W] = 0$ and $\mathbb{E}[w_\perp^2] = 1$. The value of $E$ is now $\frac{12.5}{16}$.

# C  PROOFS FOR OVERPARAMETERIZED RIDGE REGRESSION

## C.1  DIFFERENCE BETWEEN THE INFINITE ENSEMBLE AND INFINITE SINGLE MODEL.

We begin with a lemma, which shows that the prediction of kernel regressors is Lipschitz-continuous in $\lambda$ for any $x^*$ and $\lambda \geq 0$. We will denote the kernel ridge regressor with regularization parameter $\lambda$ as $h_{\infty,\lambda}^{(RR)}$, as introduced in Sec. 3.3.

**Lemma 3** (Bound on the difference between the kernel ridge regressors). *Let $\lambda, \lambda' \geq 0$ be two regularization parameters. Then, for any $x^* \in \mathcal{X}$ it holds that:*

$$|h_{\infty,\lambda'}^{(RR)}(x^*) - h_{\infty,\lambda}^{(RR)}(x^*)| \leq \sqrt{n} \cdot C_1 \cdot |\lambda' - \lambda| \cdot \sqrt{y^T K^{-4} y}$$

*where we assume $k(x_i, x^*) \leq C_1$ for all $i \in [N]$.*

*Proof.* We can write the kernel ridge regressors as $h_{\infty,\lambda}^{(RR)}(x^*) = \sum_{i=1}^n \alpha_{1,i} k(x_i, x^*)$ and $h_{\infty,\lambda'}^{(RR)}(x^*) = \sum_{i=1}^n \alpha_{2,i} k(x_i, x^*)$ with coefficients $\alpha_1$ and $\alpha_2$ given by:

$$\alpha_1 = (K + \lambda I)^{-1} y$$
$$\alpha_2 = (K + \lambda' I)^{-1} y$$

We now write $y$ in the orthonormal basis of the eigenvectors of $K$, i.e. $y = \sum_{i=1}^n a_i v_i$. We call the corresponding eigenvalues of $K$ $d_1, \ldots, d_n > 0$.

The matrix $(K + \lambda I)^{-1}$ has the same eigenvectors as $K$ and the eigenvalues are $0 < \tilde{d}_i = \frac{1}{d_i + \lambda} \leq \frac{1}{\lambda}$. Thus, we can write $\alpha_1 = \sum_{i=1}^n a_i \frac{1}{d_i + \lambda} v_i$ and $\alpha_2 = \sum_{i=1}^n a_i \frac{1}{d_i + \lambda'} v_i$.

In the next step, we bound $\|\alpha_1 - \alpha_2\|_2^2$: Using the orthonormality of the eigenvectors, we get:

$$\|\alpha_1 - \alpha_2\|_2^2 = \sum_{i=1}^n \left(a_i \left(\frac{1}{d_i + \lambda} - \frac{1}{d_i + \lambda'}\right)\right)^2$$

Now we bound $\left|\frac{1}{\lambda + d_i} - \frac{1}{\lambda' + d_i}\right| \leq \left|\frac{\lambda' - \lambda}{\lambda \lambda' + (\lambda + \lambda') d_i + d_i^2}\right| \leq \frac{|\lambda' - \lambda|}{d_i^2}$ which gives us:

$$\|\alpha_1 - \alpha_2\|_2^2 \leq \sum_{i=1}^n \left(\frac{a_i |\lambda' - \lambda|}{d_i^2}\right)^2 \leq |\lambda' - \lambda|^2 y^T K^{-4} y$$

Using this result, we can bound the difference between the predictions of the two kernel regressors at a single point $x^*$:

$$|h_{\infty,\lambda}^{(RR)}(x^*) - h_{\infty,\lambda'}^{(RR)}(x^*)| = |\sum_{i=1}^n (\alpha_{1,i} - \alpha_{2,i}) k(x_i, x^*)| \leq \sum_{i=1}^n |\alpha_{1,i} - \alpha_{2,i}| k(x_i, x^*)$$

Since $k(x_i, x^*) \leq C_1$, we get (using the relation between the 1-norm and the 2-norm):

$$|f_\lambda(x^*) - f_{\lambda'}(x^*)| \leq C_1 \sum_{i=1}^n |\alpha_{1,i} - \alpha_{2,i}| \leq C_1 \|\alpha_1 - \alpha_2\|_2 \sqrt{n} \leq \sqrt{n} \cdot C_1 \cdot |\lambda' - \lambda| \cdot \sqrt{y^\top K^{-4} y}$$

$\square$

Using similar arguments, we now show that the expected prediction of RF regressors, i.e., the prediction of the infinite ensemble of RF regressors, is Lipschitz-continuous for any $x^*$ and $\lambda \geq 0$:

**Lemma 4** (Bound on the difference between expected RF Regressors). *Under Assumption 1 and Assumption 2, the expected value of the prediction of RF regressors is Lipschitz-continuous in $\lambda$ for any $x^*$ and $\lambda \geq 0$, i.e., for any $x^*$ it holds that:*

$$|\bar{h}_{\infty,\lambda'}^{(RR)}(x^*) - \bar{h}_{\infty,\lambda}^{(RR)}(x^*)| \leq \|c^\top R^{-\top}\|\|y\|DC_2\,|\lambda' - \lambda|$$

*where $C_2$ is a constant depending on the distribution of $\Phi$.*

*Proof.* We use the characterization of $\bar{h}_{\infty,\lambda}^{(RR)}(x^*)$ from Eq. (8), which gives us the difference as

$$\left| c^\top \mathbb{E}_{W,w_\perp}\left[ WW^\top \left( \left(WW^\top + D \cdot \lambda' \cdot R^{-\top}R^{-1}\right)^{-1} - \left(WW^\top + D \cdot \lambda' \cdot R^{-\top}R^{-1}\right)^{-1}\right)\right] R^{-\top}y \right|.$$

We can now reverse some steps we made to get this characterization and write it in terms of $\Phi$ again:

$$\left| c^\top R^{-\top} \mathbb{E}_{\mathcal{W}}\left[ \Phi_{\mathcal{W}}\Phi_{\mathcal{W}}^\top \left( \left(\Phi_{\mathcal{W}}\Phi_{\mathcal{W}}^\top + D \cdot \lambda' \cdot I\right)^{-1} - \left(\Phi_{\mathcal{W}}\Phi_{\mathcal{W}}^\top + D \cdot \lambda \cdot I\right)^{-1}\right)\right] y \right|.$$

And now, using Jensen's inequality and the convexity of the two-norm, we can pull out the expected value to the outside of the difference:

$$\|c^\top R^{-\top}\| \cdot \mathbb{E}_{\mathcal{W}}\left[ \|\Phi_{\mathcal{W}}\Phi_{\mathcal{W}}^\top \left( \left(\Phi_{\mathcal{W}}\Phi_{\mathcal{W}}^\top + D \cdot \lambda' \cdot I\right)^{-1} - \left(\Phi_{\mathcal{W}}\Phi_{\mathcal{W}}^\top + D \cdot \lambda \cdot I\right)^{-1}\right) y \|\right].$$

Similarly to the proof of Lemma 3, we can write $y$ in the orthonormal basis of the eigenvectors of $\Phi\Phi^\top$ (note that we drop the subscript $\mathcal{W}$ for notational simplicity), i.e. $y = \sum_{i=1}^n a_i v_i$. Furthermore we define the eigenvalues of $\Phi\Phi^\top$ as $d_1, \ldots, d_n > 0$. The matrix $(\Phi\Phi^\top + D \cdot \lambda I)^{-1}$ again has the same eigenvectors as $\Phi\Phi^\top$ and the eigenvalues are $0 < \frac{1}{d_i + D\cdot\lambda} \leq \frac{1}{D\cdot\lambda}$.

Multiplying $y$ with $\Phi\Phi^\top(\Phi\Phi^\top + D \cdot \lambda I)^{-1}$ and $\Phi\Phi^\top(\Phi\Phi^\top + D \cdot \lambda' I)^{-1}$ then gives us:

$$\Phi\Phi^\top(\Phi\Phi^\top + D \cdot \lambda I)^{-1}y = \sum_{i=1}^n a_i \frac{d_i}{d_i + D\cdot\lambda} v_i$$

$$\Phi\Phi^\top(\Phi\Phi^\top + D \cdot \lambda' I)^{-1}y = \sum_{i=1}^n a_i \frac{d_i}{d_i + D\cdot\lambda'} v_i$$

We can now calculate the difference of these two vectors using the orthonormality of the eigenvectors:

$$\|\Phi\Phi^\top(\Phi\Phi^\top + D \cdot \lambda' I)^{-1}y - \Phi\Phi^\top(\Phi\Phi^\top + D \cdot \lambda I)^{-1}y\|_2^2 = \sum_{i=1}^n \left( a_i \left( \frac{d_i}{d_i + D\cdot\lambda} - \frac{d_i}{d_i + D\cdot\lambda'}\right)\right)^2$$

Now we look at the difference between the two coefficients and see that for each $i$, we have:

$$\left| \frac{d_i}{d_i + D\cdot\lambda} - \frac{d_i}{d_i + D\cdot\lambda'} \right| \leq \frac{D\cdot|\lambda' - \lambda|}{d_i}$$

Thus, we have that the difference is bounded by:

$$\|\Phi\Phi^\top(\Phi\Phi^\top + D \cdot \lambda' I)^{-1}y - \Phi\Phi^\top(\Phi\Phi^\top + D \cdot \lambda I)^{-1}y\|_2^2 \leq \frac{D^2 \cdot |\lambda - \lambda'|^2}{d_N^2}\|y\|_2^2.$$

All together, we can now bound the difference of the expected values of the predictions of RF regressors via:

$$|\bar{h}_{\infty,\lambda'}^{(RR)}(x^*) - \bar{h}_{\infty,\lambda}^{(RR)}(x^*)| \leq \|c^\top R^{-\top}\|\|y\|D|\lambda' - \lambda|\mathbb{E}_{d_N}\left[\frac{1}{d_N}\right]$$

Since $\text{tr}((\Phi\Phi^\top)^{-1}) = \sum_{i=1}^n \frac{1}{d_i}$, and the trace is a linear operator, we can write:

$$\mathbb{E}_{d_N}\left[\frac{1}{d_N}\right] \leq \mathbb{E}_{\mathcal{W}}\left[(\text{tr}(\Phi_{\mathcal{W}}\Phi_{\mathcal{W}}^\top)^{-1})\right] = \text{tr}(\mathbb{E}_{\mathcal{W}}\left[(\Phi_{\mathcal{W}}\Phi_{\mathcal{W}}^\top)^{-1}\right]) =: C_2$$

which is finite whenever $\mathbb{E}_{\mathcal{W}}\left[(\Phi_{\mathcal{W}}\Phi_{\mathcal{W}}^\top)^{-1}\right]$ is finite, i.e. Assumption 2 holds. $\square$

Using Lemma 3 and Lemma 4 we can now show that the difference between the infinite ensemble where each model has ridge $\lambda$ and the infinite single model with ridge $\lambda$ is Lipschtiz-continuous in $\lambda$ for $\lambda \geq 0$:

**Theorem 2 (Restated).** *Under Assumptions 1 and 2, the difference $|\bar{h}_{\infty,\lambda}^{(RR)}(x^*) - h_{\infty,\lambda}^{(RR)}(x^*)|$ between the infinite ensemble and the single infinite-width model trained with ridge $\lambda$ is Lipschitz-continuous in $\lambda$ for $\lambda \geq 0$. The Lipschitz constant is independent of $x^*$ for compact $\mathcal{X}$.*

*Proof.* We bound difference $\left| |\bar{h}_{\infty,\lambda'}^{(RR)}(x^*) - h_{\infty,\lambda'}^{(RR)}(x^*)| - |\bar{h}_{\infty,\lambda}^{(RR)}(x^*) - h_{\infty,\lambda}^{(RR)}(x^*)| \right|$ by using first the inverse, then the normal triangle inequality:

$$
\left| |\bar{h}_{\infty,\lambda'}^{(RR)}(x^*) - h_{\infty,\lambda'}^{(RR)}(x^*)| - |\bar{h}_{\infty,\lambda}^{(RR)}(x^*) - h_{\infty,\lambda}^{(RR)}(x^*)| \right|
$$
$$
\leq |\bar{h}_{\infty,\lambda'}^{(RR)}(x^*) - \bar{h}_{\infty,\lambda}^{(RR)}(x^*) + h_{\infty,\lambda}^{(RR)}(x^*) - h_{\infty,\lambda'}^{(RR)}(x^*)|
$$
$$
\leq |\bar{h}_{\infty,\lambda'}^{(RR)}(x^*) - \bar{h}_{\infty,\lambda}^{(RR)}(x^*)| + |h_{\infty,\lambda}^{(RR)}(x^*) - h_{\infty,\lambda'}^{(RR)}(x^*)|
$$

Using the bound from Lemma 3 and Lemma 4 (and summarizing the the corresponding constants as $c_1$ and $c_2$) we can bound this by:

$$
|\bar{h}_{\infty,\lambda'}^{(RR)}(x^*) - h_{\infty,\lambda'}^{(RR)}(x^*)| - |\bar{h}_{\infty,\lambda}^{(RR)}(x^*) - h_{\infty,\lambda}^{(RR)}(x^*)| \leq c_1|\lambda' - \lambda| + c_2|\lambda' - \lambda|
$$

Thus we have Lipschitz-continuity in $\lambda$ for $\lambda \geq 0$.

The Lipschitz constant is independent of $x^*$ for $\mathcal{X}$ compact since the Lipschitz constants from Lemma 3 and Lemma 4 depend on $x^*$ in a continuous fashion. $\qquad\square$

Note that an equivalent argument in combination with Jacot et al. (2020)[Proposition 4.2], i.e. $\tilde{\lambda} \leq \frac{\gamma}{\gamma-1}\lambda$, directly gives the Lipschitz-continuity in $\lambda$ for $\lambda \geq 0$ for the difference between the infinite ensemble and the infinite-width single model with effective ridge in the overparameterized regime.

# D UNDERPARAMETERIZED ENSEMBLES

Here, we offer a proof that infinite, unregularized, underparameterized RF ensembles are equivalent to kernel ridge regression under a transformed kernel function. We emphasize the difference from the overparameterized case—the central focus of our paper—in which the infinite ensemble is equivalent to a ridgeless kernel regressor. Thus, underparameterized ensembles induce regularization, while overparameterized ensembles do not.

Other works have explored the ridge behavior of underparameterized RF ensembles (Kabán, 2014; Thanei et al., 2017; Bach, 2024a); however, these works often focus on an equivalence in generalization error whereas we establish a pointwise equivalence. To the best of our knowledge, the following result is novel:

**Lemma 5.** *If the expected orthogonal projection matrix $\mathbb{E}_{\tilde{W}}\left[R^\top\tilde{W}\left(W^\top RR^\top W\right)^{-1}\tilde{W}^\top R\right]$ is well defined, and a contraction (i.e., singular values strictly less than 1), then the infinite underparameterized RF ensemble $\bar{h}_\infty^{(LN)}(x^*)$ is equivalent to kernel ridge regression under some kernel function $\tilde{k}(\cdot,\cdot)$.*

*Proof.* When $D < N$, the infinite ridgeless RF ensemble is given by

$$
\bar{h}_\infty^{(LN)}(x^*) = \mathbb{E}_\mathcal{W}\left[\frac{1}{D}\sum_{j=1}^D \phi(\omega_j, x^*)\left(\frac{1}{D}\Phi_\mathcal{W}^\top\Phi_\mathcal{W}\right)^{-1}\Phi_\mathcal{W}^\top\right]y
$$
$$
= \mathbb{E}_{W,w_\perp}\left[\left(r_\perp w_\perp^\top + c^\top W\right)\left(W^\top RR^\top W\right)^{-1}W^\top\right]Ry, \tag{11}
$$

where $W, w_\perp, r_\perp, c, R$ are as defined in Sec. 2. Defining the following block matrices:

$$
\tilde{W} = \begin{bmatrix} W \\ w_\perp^\top \end{bmatrix} \in \mathbb{R}^{(N+1)\times D}, \qquad \tilde{R} = \begin{bmatrix} R \\ 0 \end{bmatrix} \in \mathbb{R}^{(N+1)\times N}, \qquad \tilde{c} = \begin{bmatrix} c \\ r_\perp \end{bmatrix} \in \mathbb{R}^{(N+1)},
$$

we can rewrite Eq. (11) as

$$\bar{h}_\infty^{(LN)}(x^*) = \tilde{c}^\top \left( \mathbb{E}_{\tilde{W}} \left[ \tilde{W} \left( W^\top RR^\top W \right)^{-1} \tilde{W}^\top \right] \right) \tilde{R}y.$$

By adding and subtracting $\tilde{R}\tilde{R}^\top$ inside the outer parenthesis, we can massage this expression into kernel ridge regression in a transformed coordinate system:

$$\bar{h}_\infty^{(LN)}(x^*) = \tilde{c}^\top \left( \tilde{R}\tilde{R}^\top + \underbrace{\left( \mathbb{E}_{\tilde{W}} \left[ \tilde{W} \left( W^\top RR^\top W \right)^{-1} \tilde{W}^\top \right] \right)^{-1} - \tilde{R}\tilde{R}^\top}_{:=\tilde{A}} \right)^{-1} \tilde{R}y.$$

$$= \tilde{c}^\top \tilde{A}^{-1} \tilde{R} \left( \tilde{R}^\top \tilde{A}^{-1} \tilde{R} + I \right)^{-1} y. \tag{12}$$

Applying the Woodbury inversion lemma to $\tilde{A}^{-1}$, we have:

$$\tilde{A}^{-1} = \mathbb{E}_{\tilde{W}} \left[ \tilde{W} \left( W^\top RR^\top W \right)^{-1} \tilde{W}^\top \right]$$
$$+ \mathbb{E}_{\tilde{W}} \left[ \tilde{W} \left( W^\top RR^\top W \right)^{-1} W^\top R \right] \left( I - \mathbb{E}_W[P_W] \right)^{-1} \mathbb{E}_{\tilde{W}} \left[ R^\top W \left( W^\top RR^\top W \right)^{-1} \tilde{W}^\top \right], \tag{13}$$

where $P_W$ is the (random) orthogonal projection matrix onto the span of the columns of $R^\top W$:

$$P_W = R^\top W \left( W^\top RR^\top W \right)^{-1} W^\top R.$$

Because $P_W$ is an orthogonal projection matrix, we have that $\|P_W\|_2 = 1$, and thus (by Jensen's inequality) $\|\mathbb{E}_W[P_W]\|_2 \le 1$. If this inequality is strict so that $I - \mathbb{E}_W[P_W]$ is invertible, we have by inspection of Eq. (13) that $\tilde{A}$ is positive definite. Therefore, the block matrix

$$\begin{bmatrix} \tilde{R}^\top \\ \tilde{c}^\top \end{bmatrix} \tilde{A}^{-1} \begin{bmatrix} \tilde{R} & \tilde{c} \end{bmatrix} = \begin{bmatrix} \tilde{R}^\top \tilde{A}^{-1} \tilde{R} & \tilde{R}^\top \tilde{A}^{-1} \tilde{c} \\ \tilde{c}^\top \tilde{A}^{-1} \tilde{R} & \tilde{c}^\top \tilde{A}^{-1} \tilde{c} \end{bmatrix} \tag{14}$$

is also positive definite and thus the realization of some kernel function $\tilde{k}(\cdot, \cdot)$; i.e.

$$\begin{bmatrix} \tilde{R}^\top \tilde{A}^{-1} \tilde{R} & \tilde{R}^\top \tilde{A}^{-1} \tilde{c} \\ \tilde{c}^\top \tilde{A}^{-1} \tilde{R} & \tilde{c}^\top \tilde{A}^{-1} \tilde{c} \end{bmatrix} = \begin{bmatrix} \tilde{k}(x_1, x_1) & \cdots & \tilde{k}(x_1, x_N) & \tilde{k}(x_1, x^*) \\ \vdots & \ddots & \vdots & \vdots \\ \tilde{k}(x_N, x_1) & \cdots & \tilde{k}(x_N, x_N) & \tilde{k}(x_N, x^*) \\ \tilde{k}(x^*, x_1) & \cdots & \tilde{k}(x^*, x_N) & \tilde{k}(x^*, x^*) \end{bmatrix}.$$

Note that if $\tilde{A} = I$ then by Eq. (1) we recover the original kernel matrix

$$\begin{bmatrix} \tilde{R}^\top \tilde{R} & \tilde{R}^\top \tilde{c} \\ \tilde{c}^\top \tilde{R} & \tilde{c}^\top \tilde{c} \end{bmatrix} = \begin{bmatrix} k(x_1, x_1) & \cdots & k(x_1, x_N) & k(x_1, x^*) \\ \vdots & \ddots & \vdots & \vdots \\ k(x_N, x_1) & \cdots & k(x_N, x_N) & k(x_N, x^*) \\ k(x^*, x_1) & \cdots & k(x^*, x_N) & k(x^*, x^*) \end{bmatrix}.$$

Thus, the underparameterized ensemble in Eq. (12) simplifies to

$$\bar{h}_\infty^{(LN)}(x^*) = \begin{bmatrix} \tilde{k}(x^*, x_1) & \cdots & \tilde{k}(x^*, x_N) \end{bmatrix} \left( \begin{bmatrix} \tilde{k}(x_1, x_1) & \cdots & \tilde{k}(x_1, x_N) \\ \vdots & \ddots & \vdots \\ \tilde{k}(x_N, x_1) & \cdots & \tilde{k}(x_N, x_N) \end{bmatrix} + I \right)^{-1} y,$$

which is kernel ridge regression with respect to the kernel $\tilde{k}(\cdot, \cdot)$. $\qquad \square$

