# OpenReview forum: "Theoretical Limitations of Ensembles in the Age of Overparameterization"
_ICLR.cc/2025/Conference — Submitted to ICLR 2025_

### Official Review · Reviewer_NANc · 2024-11-01

**Soundness:** 2
**Presentation:** 3
**Contribution:** 2
**Rating:** 3
**Confidence:** 4

**Summary:**

This paper provided an asymptotic theoretical analysis for model ensemble in over-parameterization regime.

**Strengths:**

1. The story is clear, and the paper is well-organized.
2. The figures verify the theoretical results, and make the statement persuasive.

**Weaknesses:**

1. The result in Theorem 1 (question 1) has been proposed in a work titled "On the Benefits of Over-parameterization for Out-of-Distribution Generalization"(see [1]), which has been posted on March 2024. They have analysed such phenomenon under the same setting. Please add the citation, as well as comparing the difference between your result and the result in this work.
2. Assumption 2 might related to the distribution on data x, please characterise it with respect to x, to make the assumption more clear.

[1] Hao Y, Lin Y, Zou D, et al. On the Benefits of Over-parameterization for Out-of-Distribution Generalization[J]. arXiv preprint arXiv:2403.17592, 2024.

**Questions:**

See weaknesses.

---

> ### Author Response · Authors · 2024-11-20
>
> We thank the reviewer for their comments and feedback on our paper. Below, we address the concerns and questions raised, and outline the changes made in response.
>
> **1\. Comparison to Hao et al. (2024):**
> > The result in Theorem 1 (question 1\) has been proposed in a work titled "On the Benefits of Over-parameterization for Out-of-Distribution Generalization"(see \[1\]), which has been posted on March 2024\. They have analysed such phenomenon under the same setting. Please add the citation, as well as comparing the difference between your result and the result in this work.
>
> We thank the reviewer for bringing \[1\] to our attention. However, we respectfully but strongly disagree that our Theorem 1 was proposed by \[1\]. Significant differences exist between our work and \[1\], particularly regarding the assumptions and the nature of results, leading to very different potential for application.
>
> 1. **Assumptions:**
> - Our Theorem 1 relies on minimal assumptions, specifically subexponential tails for $w\_i w\_{\\perp i} $ (Assumption 1), and is independent of both data distributions and test-time perturbations.
> - In contrast, \[1\] assumes benign overfitting, Gaussian weights ( $W \\sim N(0,1 / p)$ ), and ReLU activations. Their results are further restricted to specific types of perturbations to the training distribution.
>
> 2\.    **Nature of results:**
>
> - Theorem 1 shows asymptotic **pointwise equivalence** between infinite ensembles and single infinite-width models without assuming Gaussianity. This equivalence serves as a foundation for analyzing ensemble variance and finite-width models.
> - \[1\] provides **high probability, non-asymptotic lower bounds** on improvements to the OOD risk via ensembling and increasing model capacity. While both ensembling and capacity increases improve OOD risk in their setting, \[1\] does not analyze whether these improvements are equivalent in the limit or if their rates are similar (as they only give lower bounds).
>
> **Changes:**
>
> - We added a citation to \[1\] and a discussion of its relationship to our results in the related work section.
>
> **2\. Clarification of Assumption 2:**
>
> >Assumption 2 might relate to the distribution on data $x$, please characterize it with respect to $x$, to make the assumption more clear.
>
> In our work, we assume a fixed training dataset (as described in Section 2), and it is typically assumed that the data matrix $X$ has full rank. Under this assumption, the distribution of weights and the specific activation function primarily determine whether Assumption 2 holds.
> Notably, Assumption 2 is directly fulfilled under the Gaussian universality assumption. In this case, $(\\Phi\_{\\mathcal{W}} \\Phi\_{\\mathcal{W}}^\\top )^{-1}$ follows an inverse Wishart distribution, and $\\mathbb{E}\[(\\Phi\_{\\mathcal{W}} \\Phi\_{\\mathcal{W}}^\\top)^{-1}]$ is always finite (see [Inverse Wishart Distribution](https://en.wikipedia.org/wiki/Inverse-Wishart_distribution)). We expect that Assumption 2 is similarly satisfied for many other relevant distributions of $\\Phi\_{\\mathcal{W}}$, but a comprehensive characterization for non-Gaussian distributions remains challenging.
>
> ---
>
> We hope that these responses address your concerns and clarify the contributions of our work. If you have further questions or suggestions, we would be happy to address them.
>
> ---
>
> [1] Hao Y, Lin Y, Zou D, et al. On the Benefits of Over-parameterization for Out-of-Distribution Generalization[J]. arXiv preprint arXiv:2403.17592, 2024.

---

### Official Review · Reviewer_2wzE · 2024-11-06

**Soundness:** 3
**Presentation:** 3
**Contribution:** 2
**Rating:** 5
**Confidence:** 3

**Summary:**

This paper explores the generalization and uncertainty quantification aspects of overparameterized ensembles, particularly in neural networks. It examines how modern overparameterized ensembles, such as those constructed from neural networks, differ from traditional, underparameterized ensembles. Using random feature (RF) regressors, the paper theoretically demonstrates that overparameterized ensembles tend to be equivalent to a single large model in terms of generalization and predictive variance. Key contributions include proving that, under certain conditions, overparameterized ensembles provide no inherent generalization advantage over a single large model, and showing that predictive variance in overparameterized ensembles does not equate to traditional uncertainty measures.

**Strengths:**

- Builds on existing theoretical framework to address an important question of whether there is any advantage of training ensembles vs a bigger model.
- The assumptions are clearly stated and the proofs of the theorems are complete.
- The notation and setup are clearly explained

**Weaknesses:**

(See Questions for details)
- The paper does not explore empirical implications of the theoretical results
- The results are preliminary in nature - focus mostly on asymptotic regime and corresponding finite-regime rates are not provided
- Very few examples to illustrate the assumptions and the actual implications of the theorems.

**Questions:**

- Assumption 1.2 (a.s. Positive definite): Can you provide some examples of features distributions for which this assumption is true. For the ReLU and Leaky-ReLU approximations, can you provide the precise rate at which the approximations yield the positive definite matrix. For example, in footnote (2), what is the dependence on $\alpha$ in the condition number of the matrix \sum_i w_i w_i^\top?
- Theorem 1: Could you please explain where in the proof of this theorem would you need the assumption that D >> N? What is the rate of convergence in terms of number of elements in the ensembles and in terms of the number of features of the single model?
- Theorem 1: What are the main challenges in obtaining a finite dimensional version of this theorem?
- Theorem 1: Can you please instantiate the rates of the theorem for a few different classes of feature vectors which satisfy Assumption 1?
- What classes of features and distributions satisfy assumption 2?
- Theorem 2: what is the exact characterization of the Lipschitz constant \lambda? Can you provide a few examples for this constant for different distributions/features?
- What are the implications of these results for actual deep networks? How well do these results transfer empirically to the deep learning regime? It would be useful to provide some experiments with small real-world datasets like Cifar-10 to study the implications of these results.

---

> ### Author Response · Authors · 2024-11-20
>
> We thank the reviewer for their comments and valuable feedback on our paper. Below, we address the concerns and questions raised and state the changes we have made in response.
>
> **1\. Assumption 1.2 (a.s. Positive Definite):**
> > Can you provide some examples of feature distributions for which this assumption is true?
>
> If $\\mathcal{X}$ is of full rank, the weights are sampled i.i.d. from $\\mathcal{N}(0, 1)$, and the data dimensionality satisfies $p \\geq D, N$, the feature matrix $\\Phi\_{\\mathcal{W}}$ is almost surely full rank when using Leaky ReLU as the activation function. We also expect most nonlinear activation functions, such as sigmoid, tanh, sine, or cosine, to satisfy this condition with i.i.d. weights from a distribution with a density function. However, rigorously proving this is beyond the scope of this work.
>
> **2\. Rate of Convergence for ReLU and Leaky-ReLU Approximations:**
> > For the ReLU and Leaky-ReLU approximations, can you provide the precise rate at which the approximations yield the positive definite matrix? For example, in footnote (2), what is the dependence on the condition number of the matrix $\\sum\_i w\_i w\_i^\\top$?
>
> We are not entirely sure what you refer to by "rates." Are you referring to the convergence speed of the approximations:
> \\\[
> \\phi\_\\alpha\\left(\\omega^{\\top} x\\right) \= \\frac{1}{\\alpha} \\log \\left(1 \+ e^{\\alpha \\omega^{\\top} x}\\right),
> \\\]
> to \\(\\max(0, \\omega^\\top x)\\) as \\(\\alpha\\) increases?
>
> If so, the rate at which the difference \\(\\phi\_\\alpha(\\omega^\\top x) \- \\max(0, \\omega^\\top x)\\) decays is typically exponential in $\\alpha$.
>
> **3\. Theorem 1: Assumption (D \> N):**
> > Could you please explain where in the proof of this theorem would you need the assumption that (D \> N)?
>
> The expression for the optimal weights differs between the two regimes:
> - **Overparameterized case $(D \> N)$:** $\\theta \= \\Phi^\\top (\\Phi \\Phi^\\top)^{-1} y$
> - **Underparameterized case $(D \\leq N)$**: $\\theta \= (\\Phi^\\top \\Phi)^{-1} \\Phi^\\top y$
>
> This distinction leads to differences in the ensemble predictions. In the **overparameterized regime**, the expansion for the ensemble prediction is
> \\[
> \begin{align*}
> \\overline{h}\_\\infty(x^*) &= \\mathbb{E}\_{\\mathcal{W}} \\left\[\\tfrac{1}{D} \\phi\_\\mathcal{W}^* \\Phi\_\\mathcal{W}^\\top \\left(\\tfrac{1}{D} \\cdot \\Phi\_\\mathcal{W} \\Phi\_\\mathcal{W}^\\top\\right)^{-1}\\right\] y, \\\\
> &= \\mathbb{E}\_{W, w\_\\bot} \\left\[\\left(c^\\top W \+ r\_\\bot w\_\\bot^\\top\\right) W^\\top R \\left(R^\\top W W^\\top R\\right)^{-1}\\right\] y.
> \end{align*}
> \\]
>
> Here, $R R^\\top$ can be factored out of the inverse because it surrounds $W W^\\top$, enabling the required simplifications (see the proof of equation (2) in the appendix).
>
> In the **underparameterized regime**, the expansion is instead:
> \\[
> \begin{align*}
> \\overline{h}\_\\infty(x^*) &= \\mathbb{E}\_{\\mathcal{W}} \\left\[\\phi\_\\mathcal{W}^* \\left(\\Phi\_\\mathcal{W}^\\top \\Phi\_\\mathcal{W}\\right)^{-1} \\Phi\_\\mathcal{W}^\\top\\right\] y \\\\
> &= \\mathbb{E}\_{W, w\_\\bot} \\left\[\\left(c^\\top W \+ r\_\\bot w\_\\bot^\\top\\right) \\left(W^\\top R R^\\top W\\right)^{-1} W^\\top R\\right\] y.
> \end{align*}
> \\]
>
> The key difference is that in the underparameterized case, $R R^\\top$ lies between $W$ and $W^\\top$ within the inverse, preventing it from being factored out. We empirically demonstrate in Figure 3 that the equivalence in Theorem 1 fails in the underparameterized regime. Lastly, we emphasize that the assumption $D \> N)$ (not $D \\gg N)$) is sufficient for the results in Theorem 1\.
>
> **Changes:**
> - We clarified the role of (D \> N) below the proof of equation (2) in the appendix.
>
> **4\. Rate of Convergence for Ensembles and Single Models:**
> >What is the rate of convergence in terms of the number of elements in the ensembles and the number of features in a single model?
>
> - **Ensemble Size (M):** The convergence rate in terms of the number of ensemble members is $O(1/\\sqrt{M})$, based on standard Monte Carlo sampling theory. We can use MC sampling theory since, in any finite ensemble, we effectively estimate the expected value in equation (2) using a finite number of random samples.
> - **Feature Width (D):** The variance of a random feature model prediction captures the expected squared difference between the prediction of the ensemble's limit (e.g., the infinite-width model) and that of a single finite-width model. As noted in the paper and supported by prior work (e.g., Adlam & Pennington, 2020\) and empirical results (see Fig. 5 and Appendix A.3), this variance decays as $\\sim 1 / D$ across various feature distributions. Thus, we have a convergence at the rate $O(1/\\sqrt{D})$.

---

> ### Author Response · Authors · 2024-11-20
>
> **5\. Challenges in Finite-Dimensional Versions of Theorem 1:**
> > What are the main challenges in obtaining a finite-dimensional version of this theorem?
>
> We are not sure what you mean by a "finite-dimensional version". Are you referring to an analysis of ensembles with a finite number of members? We analyze this setting in Section 3 under the heading "Ensembles versus larger single models under a finite feature budget." We apologize if we misunderstood your question.
>
> **6\. Instantiating Rates for Feature Classes:**
> > Can you please instantiate the rates of the theorem for a few different classes of feature vectors which satisfy Assumption 1?
>
> Assuming $X$ is of full rank and weights are sampled i.i.d. from $\\mathcal{N}(0,1)$, Assumption 1 is satisfied for a variety of activation functions, including sigmoid, Gaussian error function, softplus, tanh, sine, and cosine. Regarding the rates, these should generally be independent of the specific feature vector assumptions. Please refer to our response in Point 4 for a discussion of the convergence rates in terms of ensemble size $M$ and feature width $D$.
>
> **7\. Assumption 2:**
> >What classes of features and distributions satisfy Assumption 2?
>
> Assumption 2 is satisfied under the Gaussian universality assumption, where $(\\Phi\_\\mathcal{W} \\Phi\_\\mathcal{W}^\\top)^{-1}$ follows an inverse Wishart distribution. In this case, the expected value $\\mathbb{E}\[(\\Phi\_\\mathcal{W} \\Phi\_\\mathcal{W}^\\top)^{-1}\]$ is well-defined and finite (see, e.g., [Wikipedia on Inverse Wishart](https://en.wikipedia.org/wiki/Inverse-Wishart_distribution)). We also expect Assumption 2 to hold for many sub-Gaussian feature distributions, but we were unable to rigorously show this for non-Gaussian cases.
>
> Importantly, Assumption 2 is only required in Theorem 2 to allow us to analyze the transition between the positive ridge regime ($\\lambda \> 0$) and the zero-ridge regime ($\\lambda \= 0$). Without this assumption, $\\lambda \\rightarrow 0$ could not be directly investigated, i.e., we could only state Theorem 2 for $\\lambda \> \\epsilon \> 0$.
>
> **Changes:**
> - We added that Assumption 2 is strictly stronger than the Gaussian universality assumption.
>
> **8\. Lipschitz Constant in Theorem 2:**
> >What is the exact characterization of the Lipschitz constant? Can you provide examples for different distributions/features?
>
> The Lipschitz constant (C) in Theorem 2 follows from Lemmas 3 and 4 in the appendix. It is given by:
> $$
> C = \\max\\left\\{\\sqrt{n} \\cdot C_1 \\cdot \\sqrt{y^\\top K^{-4} y}, |c^\\top R^{-\\top}| \\cdot |y| \\cdot D \\cdot C_2 \\right\\}
> $$
> where $C\_1$ bounds the kernel evaluations $k(x\_i, x^\*)$ for all $i \\in \[N\]$, and $C\_2 \= \\operatorname{tr}(\\mathbb{E}\[(\\Phi\_\\mathcal{W} \\Phi\_\\mathcal{W}^\\top)^{-1}\])$.
>
> For Gaussian features, $C\_2$ could be computed explicitly as the trace of the inverse Wishart distribution's expected value. However, the Lipschitz constant is dependent on the specific training data and kernel matrix. As a result, its practical computation is pretty cumbersome and, in our opinion, not particularly informative.
>
> **8\. Implications for Deep Learning and Real-World Datasets:**
> >What are the implications of these results for actual deep networks? How well do these results transfer empirically to the deep learning regime? It would be useful to provide some experiments with small real-world datasets like Cifar-10 to study the implications of these results.
>
> The primary contribution of our paper is a theoretical analysis to explain empirical phenomena observed in recent works, including Abe et al. (2022; 2024\) and Theisen et al. (2024). These studies demonstrate that ensembles of overparameterized neural networks often behave similarly to single large models, which is consistent with the predictions of our theoretical framework. We thus refer you to these papers to see that our theoretical results predict phenomena for actual deep networks.
>
> Our experiments focused on the California Housing dataset ([https://www.kaggle.com/datasets/camnugent/california-housing-prices](https://www.kaggle.com/datasets/camnugent/california-housing-prices)) because it is a regression dataset compatible with our assumptions. In contrast, CIFAR-10 and MNIST are classification datasets, which would have required extending our theoretical framework. Additionally, CIFAR-10 and MNIST would require relatively larger sample sizes for meaningful experiments, potentially exacerbating numerical stability issues (discussed in Appendix A.2).
>
> **Changes:**
> - We more explicitly stated that our theoretical findings align with recent empirical observations in Abe et al. (2022; 2024\) and Theisen et al. (2024) in the discussion of Theorem 1 and Section 3.2.
>
> ---
>
> We hope these responses address your concerns and clarify the contributions of our work. If you have any further questions or suggestions, we would be happy to discuss them.

---

### Official Review · Reviewer_bNwj · 2024-11-09

**Soundness:** 3
**Presentation:** 3
**Contribution:** 2
**Rating:** 5
**Confidence:** 4

**Summary:**

This paper studies the effect of ensembling in random features (RF) models. The basic question in this area is whether ensembling provides any advantages over a single larger model, such as implicit regularization or better generalization. Building on the work of Jacot et al. (2020), the authors demonstrate that in the regression setting, an infinite ensemble average of ridgeless RF models is equivalent to a single infinitely-wide ridgeless RF model. Notably, this result does not rely on the gaussian universality assumption common in the literature. The authors then discuss the positive ridge case, showing that the same pointwise result holds up to an error proportional to lambda. As a consequence, they point out that popular ensemble-based uncertainty measures capture deviations from an infinite-width limit, not the true response, and therefore must be interpreted carefully.

**Strengths:**

The paper is clearly written and provides a good overview of recent papers in the area. The strongest result in the paper is Theorem 1. This theorem establishes equivalence results under Assumption 1, which assumes subexponential distributions on the whitened random features and allows for dependence. This is weaker than the gaussian process assumption required by Jacot et al (2020). The key technical result is Lemma 1, which leverages concentration inequalities for subexponential random variables to handle an error term in the pointwise analysis of Theorem 1. The experiments are clear and support Theorem 1 and its corollaries.

**Weaknesses:**

One concern is that Theorem 1 does not offer substantially new insights into ensembling compared to Jacot et al. (2020), which also showed the equivalence of ridgeless ensembles and a ridgeless infinite-width limit, albeit under stronger assumptions.
Also, I believe the paper could be strengthened if it provided more theoretical analysis in the positive ridge setting. The authors prove that the error is Lipschitz in the regularization $\lambda$ in Theorem 2. However, in Jacot et al. (2020), the authors characterize the regularization $\tilde \lambda$ of the ensemble and show that $\tilde \lambda$ is strictly greater than $\lambda$. If the results in Theorem 2 are sharpened, can this implicit regularization be recovered?
Finally, I believe the discussion of uncertainty quantification in Section 3.2 could use further development. In particular, can the results in Theorem 1 or Theorem 2 be used to demonstrate a lack of robustness to distribution shifts, as discussed in Abe et al. (2022)?

**Questions:**

Can Theorem 1 be generalized to account for feature learning? For instance, does the result hold true if 1 step of gradient descent is performed on the first layer weights? Is it possible for the analysis in Theorem 2 to be refined so as to place bounds on the implicit regularization $\tilde \lambda$ of the ensembled model? Also, can the discussion in Section 3.2 be used to evaluate robustness of the ensemble to distribution shifts?

---

> ### Author Response · Authors · 2024-11-20
>
> We thank the reviewer for their comments and valuable feedback on our paper. Below, we address the concerns and questions raised and state the changes we have made in response.
>
> **1\. Comparison to Jacot et al. (2020):**
>
> > One concern is that Theorem 1 does not offer substantially new insights into ensembling compared to Jacot et al. (2020), which also showed the equivalence of ridgeless ensembles and a ridgeless infinite-width limit, albeit under stronger assumptions.
>
> We would like to clarify why results derived under weaker, more realistic assumptions, such as those in our work, are relevant.
>
> Most importantly, weaker assumptions **reveal deeper and more universal properties of ensembling**. Our results demonstrate that the convergence of the infinite ensemble to the infinite-width model is a **fundamental property of overparameterization**, independent of Gaussianity. This underscores the findings from Jacot et al. (2020). In contrast, as we show in Section 3.2, ensemble variance aligns with Gaussian process posterior variance under the Gaussianity assumption, while it behaves differently under weaker assumptions. These differences highlight the need for caution when interpreting ensemble variance as a measure of uncertainty.
>
> Moreover, using more realistic assumptions – such as relaxing independence assumptions between activations – leads to results better aligned with the behavior of large, practical machine learning systems.
>
> **Changes:**
>
> - We expanded the discussion of the differences between our work and Jacot et al. (2020) in the related work section and added more details on the meaning of our results under these weaker assumptions in the discussion of Theorem 1\.
>
> **2\. Theoretical analysis in the positive ridge setting:**
>
> > Also, I believe the paper could be strengthened if it provided more theoretical analysis in the positive ridge setting. The authors prove that the error is Lipschitz in the regularization $\\lambda$ in Theorem 2\. However, in Jacot et al. (2020), the authors characterize the regularization $\\tilde{\\lambda}$ of the ensemble and show that $\\tilde{\\lambda}$ is strictly greater than $\\lambda$. If the results in Theorem 2 are sharpened, can this implicit regularization be recovered?
>
> Our goal in Section 3.3 was to investigate whether the pointwise equivalence between ensembles and single models approximately persists when small ridge regularization is introduced and whether the transition from the positive ridge regime to the ridgeless regime is smooth.
>
> In particular, it was a priori not clear whether the transition from the positive ridge regime to the zero ridge regime would be smooth. Even under the significantly stronger assumptions of Jacot et al. (2020), they note that the constants in their bounds explode as $\\lambda \\rightarrow 0$ (see remark on page 6). Since the assumptions by Jacot et al. (2020) include our Assumptions 1 and 2, and the implicit regularization is bounded by $\\bar{\\lambda} \\leq \\frac{\\gamma}{\\gamma-1} \\lambda$ as shown in Jacot et al. (2020), Proposition 4.2, one can generalize the proof of our Theorem 2 to the difference between a kernel model with ridge $\\tilde{\\lambda}$ and the infinite ensemble. This implies that the results from Corollary 3.2 and Theorem 4.1 from Jacot et al. (2020) are smooth in the limit as $\\lambda \\rightarrow 0$. We emphasize this point in the revision.
>
> Regarding the implicit regularization $\\tilde{\\lambda}$, Jacot et al. (2020) leverage the stronger Gaussian assumptions to characterize it explicitly. While such results would strengthen our work, we currently do not see a feasible approach without significantly reducing the generality of our assumptions.
>
> **Changes:**
>
> - We reformulated the start of Section 3.3 to clarify what we focus on in our positive ridge analysis.
> - We added a note on that our results extend the ones from Jacot et al. (2020) even in their setting when presenting Theorem 2\.

---

> ### Author Response · Authors · 2024-11-20
>
> **3\. Uncertainty quantification and robustness to distribution shifts:**
>
> > Finally, I believe the discussion of uncertainty quantification in Section 3.2 could use further development. In particular, can the results in Theorem 1 or Theorem 2 be used to demonstrate a lack of robustness to distribution shifts, as discussed in Abe et al. (2022)?
>
> Yes, our results can be used to demonstrate a lack of robustness to distribution shifts. To see this, note that Theorem 1 establishes that the infinite ensemble and the infinite-width single model are **pointwise equivalent almost surely**, implying identical generalization errors across any test distribution, including under distribution shifts.
>
> In the finite setting, as discussed in the "Ensembles versus larger single models under a finite feature budget" portion of Section 3, ensembles do not provide additional robustness compared to single models trained with the same total features, as they achieve the same bias and asymptotic variance. The argument we make here does not depend on the concrete test distribution, and thus provides general support for Abe et al.’s conclusion that "ensemble diversity does not yield additional robustness over what standard single networks achieve.”
>
> **Changes:**
>
> - We expanded the discussion in Section 3.2 slightly to explicitly connect our findings to robustness under distribution shifts, as discussed in Abe et al. (2022).
>
> **4\. Accounting for feature learning:**
>
> > Can Theorem 1 be generalized to account for feature learning? For instance, does the result hold true if 1 step of gradient descent is performed on the first layer weights?
>
> Thank you for the interesting question. A general answer to this question is out of scope for our current work, but it is certainly a direction we would be interested in exploring in the future.
>
> Intuitively, an extension of Theorem 1 to account for feature learning should likely hold under certain conditions. Specifically, if we update the first-layer weights via:
> \\\[
> \\omega\_j^{\\prime} \= \\omega\_j \- \\eta \\nabla\_{\\omega\_j} L(h),
> \\\]
> where
> \\\[
> \\nabla\_{\\omega\_j} L(h) \= \\frac{1}{\\sqrt{D}} \\theta\_j \\sum\_{i=1}^N \\left(h(x\_i) \- y\_i\\right) \\nabla\_{\\omega\_j} \\phi(\\omega\_j, x\_i),
> \\\]
> and ensure that \\(\\theta\_j\\) is initially also randomly chosen and only updated after the update of \\(\\omega\_j\\), then the updated weights \\(\\omega\_j^{\\prime}\\) still represent i.i.d. draws from a transformed version of the initial distribution.
>
> The critical question then becomes whether the updated weight distribution still satisfies Assumption 1, particularly Assumption 1.2. Intuitively, since the update likely only introduces a bounded change to the weights, we expect that Assumption 1.2 should, in many cases, remain valid if it holds for the initial distribution. However, verifying this rigorously would require further analysis.
>
> ---
>
> We hope these responses address your concerns and clarify the contributions of our work. If you have any further questions or suggestions, we would be happy to discuss them.

---

### Author Response · Authors · 2024-11-20

We thank the reviewers for their valuable feedback. Significant changes in the revised version are marked in blue, and we have addressed each comment individually. We are more than happy to elaborate on any of the answers in more detail.

---

### Author Response · Authors · 2024-12-02

As the discussion period will end soon, we kindly ask if our responses have adequately addressed your concerns and answered your questions or if there is anything additional we should provide further detail on.

---

### Meta-Review · Area_Chair_udSs · 2024-12-17

**Metareview:**

This paper studies the theoretical limitations of ensembles under overparameterization. The reviewers raised a number of concerns, including the limited innovation and insight gained by generalizing the results of  Jacot et al (2020) to more general distributions (what does one gain by generalizing to non-Gaussian? can you be specific?), and the unclear scope of some of the assumptions. Further, a closely related work by Hao et al. (2024) has not been discussed in the original submission. While the authors provided a discussion in the rebuttal, the reviewer who raised it was not able to answer, making it unclear if the discussion was complete and adequate. Given these issues, it will be better for the community if the authors revise their paper and resubmit to a future venue.

**Additional Comments On Reviewer Discussion:**

As mentioned above, the reviewers raised a number of concerns, including the limited innovation and insight gained by generalizing the results of  Jacot et al (2020) to more general distributions (what does one gain by generalizing to non-Gaussian? can you be specific?), and the unclear scope of some of the assumptions. Further, a closely related work by Hao et al. (2024) has not been discussed in the original submission. While the authors provided a discussion in the rebuttal, the reviewer who raised it was not able to answer, making it unclear if the discussion was complete and adequate. Given these issues, it will be better for the community if the authors revise their paper and resubmit to a future venue.

---

### Decision · Program_Chairs · 2025-01-22

Reject